# A Closer Look on Memorization in Tabular Diffusion Model: A Data-Centric Perspective

**Zhengyu Fang**                                                    *zxf177@case.edu*
*Department of Computer and Data Sciences*
*Case Western Reserve University*

**Zhimeng Jiang**                                                  *zhimengj@tamu.edu*
*Department of Computer Science & Engineering*
*Texas A&M University*

**Huiyuan Chen**                                                    *hxc501@case.edu*
*Department of Computer and Data Sciences*
*Case Western Reserve University*

**Xiaoge Zhang**                                                    *xxz705@case.edu*
*Department of Computer and Data Sciences*
*Case Western Reserve University*

**Kaiyu Tang**                                                      *kxt439@case.edu*
*Department of Computer and Data Sciences*
*Case Western Reserve University*

**Xiao Li**                                                          *xxl1015@case.edu*
*Department of Biochemistry*
*Center for RNA Science and Therapeutics*
*Department of Biomedical Engineering*
*Department of Computer and Data Sciences*
*Case Western Reserve University*

**Jing Li**                                                          *jingli@cwru.edu*
*Department of Computer and Data Sciences*
*Case Western Reserve University*

**Reviewed on OpenReview:** *https://openreview.net/forum?id=p2n88DfaXB*

## Abstract

Diffusion models have shown strong performance in generating high-quality tabular data, but they carry privacy risks by inadvertently reproducing exact training samples. While prior work focuses on data augmentation for memorization mitigation, little is known about which individual samples contribute the most to memorization. In this paper, we present the first *data-centric* study of *memorization dynamics* in tabular diffusion models. We begin by quantifying memorization for each real sample based on how many generated samples are flagged as its memorized replicas, using a relative distance ratio metric.[1] Our empirical analysis reveals a *heavy-tailed* distribution of memorization counts: a small subset of samples disproportionately contributes to leakage, a finding further validated through sample-removal experiments. To better understand this effect, we divide real samples into the top- and non-top-memorized groups (tags) and analyze their training-time behavior differences. We track *when* each sample is first memorized and monitor per-epoch *memorization*

---

[1] We follow Yoon et al. (2023); Gu et al. (2023); Fang et al. (2024) to define sample memorization using the relative distance ratio.

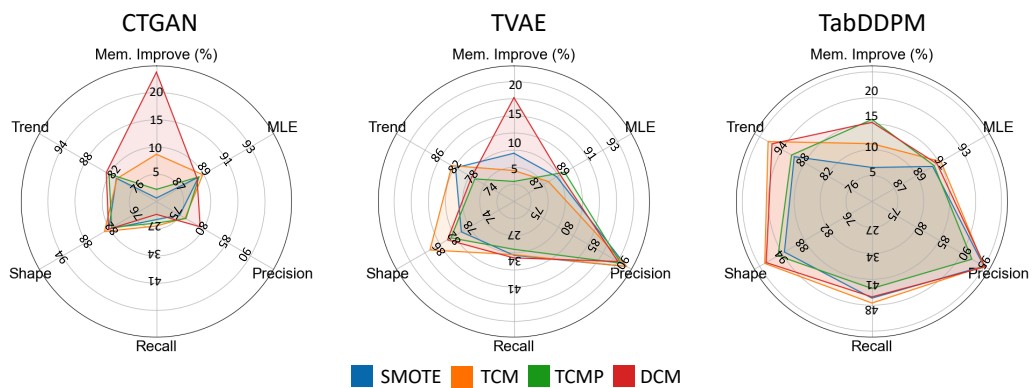

Figure 1: Overview of DynamicCutMix (DynamicCut + TabCutMix) performance in CTGAN, TVAE and TabDDPM on Adult dataset. "Mem. Improve" denotes the normalized reduction in memorization ratio. Higher values, better mitigation effectiveness.

*intensity* (AUC) across groups. We find that memorized samples tend to be memorized slightly earlier and show significantly stronger memorization signals in early training stages. Based on these insights, we propose *DynamicCut*, a two-stage, model-agnostic mitigation method. DynamicCut (a) ranks real samples by their epoch-wise memorization intensity, (b) prunes a tunable top fraction, and (c) retrains the model on the filtered dataset. Across multiple benchmark tabular datasets and tabular diffusion models, DynamicCut reduces memorization ratios with negligible impact on data diversity and downstream task performance, and complements existing data augmentation methods for further memorization mitigation. Furthermore, DynamicCut has transferability across different generative models for memorization sample tagging, i.e., high-ranked samples identified from one model (e.g., a diffusion model) are also effective in reducing memorization when removed from other generative models such as GANs and VAEs.

## 1 Introduction

Tabular data generation has gained increasing attention due to its broad applicability in healthcare, finance, and digital platforms, supporting tasks such as data augmentation, privacy preservation, and counterfactual analysis Hernandez et al. (2022); Fonseca & Bacao (2023); Assefa et al. (2020). Recent advances in generative modeling—particularly diffusion models have significantly improved the fidelity and diversity of synthetic tabular data Kotelnikov et al. (2023); Zhang et al. (2023); Yang et al. (2024); Shi et al. (2025). These models outperform traditional GAN- or VAE-based approaches Xu et al. (2019) by offering better distributional coverage and sample quality across mixed-type and high-dimensional tabular domains.

Despite the impressive performance of diffusion models in synthesizing high-quality tabular data, memorization remains a critical concern. In particular, tabular generative models risk reproducing individual training records that, while appearing structurally altered, still retain sensitive attributes of real individuals. This subtle form of privacy leakage violates core principles of generative modeling and has severe implications in sensitive domains such as healthcare and finance Karras et al. (2022); Carlini et al. (2021); Song et al. (2021); Ho et al. (2020). To address this, prior work has primarily focused on memorization detection based on nearest-neighbor distance ratios and dataset-level augmentation strategies such as feature mixing Tab-CutMix Fang et al. (2024). However, these methods fall short in two key ways. First, the memorization detection tool does not explain *when* or *which samples* are memorized during training. Second, the memorization mitigation method like TabCutMix operates uniformly across the dataset, ignoring the heterogeneous nature of sample-level memorization risks. This gap in understanding motivates a central question:

**Can we proactively identify and mitigate memorization in tabular diffusion models by monitoring sample-level training dynamics in real time?**

To answer this, we take a data-centric perspective and conduct the first fine-grained analysis of memorization dynamics in tabular diffusion models. Specifically, we seek to characterize: (1) **When** memorization arises during training; (2) **Which** samples are more likely to be memorized; (3) **How** memorization evolves and varies across examples. We show that memorization is not evenly distributed across samples: a small subset of training samples is disproportionately memorized. By tracking when each real sample is first memorized and quantifying its memorization intensity over epochs, we reveal strong early-stage signals that can be exploited to mitigate memorization.

Building on these insights, we propose a model-agnostic method for memorization mitigation, named **DynamicCut**, to identify and remove the samples with high-contribution memorization based on their early memorization intensity. DynamicCut can also be integrated into the existing data augmentation method TabCutMix, named **DynamicCutMix**, to further reduce memorization for tabular generative models. Our approach consistently reduces memorization across diverse datasets and generative models (e.g., TabDDPM, CTGAN, TVAE) without compromising data quality. Notably, the memorization mitigation benefits from samples with high-contribution memorization removal learned from one generative model can *generalize* to other generative models, demonstrating strong transferability and practicality.

## 2 Related Work

**Tabular Generative Models.** The generation of synthetic tabular data has attracted increasing attention with wide applications. Early methods such as TableGAN Park et al. (2018), CTGAN, and TVAE Xu et al. (2019) used GANs Goodfellow et al. (2020) and VAEs Kingma (2013) to address feature imbalance. Extensions like GANBLR Zhang et al. (2021) and VAE-BGM Apellániz et al. (2024) incorporate Bayesian structures to better model feature interactions and latent distributions.

To enhance dependency modeling, GOGGLE Liu et al. (2023) uses graph neural networks, while GReaT Borisov et al. (2023) reformulates tabular rows as text for table-level modeling. Other alternatives include TabPFGen Ma et al. (2024), which employs a label-weighted Energy-Based Model with in-context learning. Recently, diffusion models—originally used in image generation Ho et al. (2020)—have shown strong performance on tabular data. Notable examples include STaSy Kim et al. (2023), TabDDPM Kotelnikov et al. (2023), CoDi Lee et al. (2023), TabSyn Zhang et al. (2023), Balanced Tabular Diffusion Yang et al. (2024), and TabDiff Shi et al. (2025).

**Memorization in Generative Models.** Memorization has been widely studied in image and language generation van den Burg & Williams (2021); Gu et al. (2023); Somepalli et al. (2023b); Wen et al. (2024); Hintersdorf et al. (2024); Huang et al. (2024); Shah et al. (2025), where models often replicate training data. In vision, diffusion models like Stable Diffusion Rombach et al. (2022) and DDPM Ho et al. (2020) exhibit memorization Somepalli et al. (2023a); Carlini et al. (2021), mitigated by methods such as concept ablation Kumari et al. (2023) and adaptive sample suppression Chen et al. (2024). In language models, techniques like Goldfish Loss Hans et al. (2024) and early memorization prediction Biderman et al. (2024) aim to reduce overfitting to specific sequences. However, memorization in *tabular* data generation remains underexplored. Due to the structured and mixed-type nature of tabular data, understanding how memorization arises—and how it can be mitigated—presents a distinct and open research challenge.

## 3 Preliminary

### 3.1 Long-Tailed Distribution of Memorization Count

To understand the memorization behavior of diffusion-based tabular models, we conducted an empirical analysis using TabDDPM Kotelnikov et al. (2023) as a representative example. For each real training sample, we quantified sample-level memorization by counting the number of generated samples flagged as its replicas, based on the relative distance ratio criterion.

**Obs.1: Long-tailed Memorization Distribution.** Figure 2 reveals that memorization frequency follows a highly skewed, long-tail distribution. Most training samples are rarely recalled, whereas a small subset is

Table 1: Comparison of memorization ratio after removing different subsets of training samples on the Adult dataset.

| Dataset | Method | Mem. Ratio (%) ↓ | Improv. |
|---------|--------|------------------|---------|
| Adult | TabDDPM | 31.33 | — |
| | -Random 5% | 31.82 | **−1.56% ↓** |
| | -Label 5% | 26.12 | **16.63% ↓** |
| | -Random 10% | 26.35 | **15.90% ↓** |
| | -Label 10% | 19.35 | **38.24% ↓** |
| | -Random 20% | 27.63 | **11.81% ↓** |
| | -Label 20% | 19.86 | **36.61% ↓** |

repeatedly memorized across generations. This uneven distribution highlights that diffusion models tend to favor specific samples during generation, reflecting an inherent imbalance in how training data are retained.

**Obs.2: Influence of High-frequency Samples.** A small number of samples dominate the overall memorization behavior, suggesting that certain instances exert disproportionate influence on model recall. Identifying and analyzing these high-frequency samples is key to understanding the origin of memorization bias and evaluating its consequences for both generation quality and potential privacy leakage.

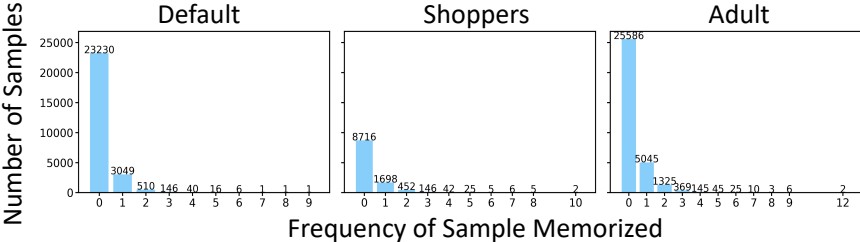

Figure 2: Memorization frequency per training sample shows a long-tail distribution, where a small number of samples are memorized much more frequently than others.

### 3.2 Tagging High-Memorization Samples

Building on the observed long-tailed distribution of sample-level memorization, we assign binary tags to training samples based on how frequently they are memorized during generation. Specifically, we rank all samples by their memorization count and define three threshold levels: the top 5%, top 10%, and top 20% most-memorized samples. Each sample is then tagged to indicate whether it falls within these high-risk subsets. These tags serve as the basis for analyzing memorization dynamics and designing targeted interventions.

### 3.3 Effectiveness of Targeted Sample Removal

To assess the practical impact of these high-memorization samples, we compare two intervention strategies: (1) randomly removing a specified portion of the training data, and (2) selectively removing the most-memorized samples as determined by our tagging. Table 1 presents memorization ratios for both strategies on the ADULT dataset.

**Obs.1: Effect of Targeted Sample Removal.** Removing high-memorization samples consistently reduces overall memorization more effectively than random removal. For instance, eliminating 10% of samples at random results in a 15.90% reduction, whereas excluding the top 10% most-memorized samples yields a 38.24% reduction. This pattern persists across all removal levels (5%, 10%, 20%), demonstrating the disproportionate impact of a small subset of memorized data.

**Obs.2: Concentration and Mitigation of Memorization.** Memorization is concentrated within a limited subset of training data, implying that modest yet targeted interventions can substantially reduce memorization without large-scale data pruning. This finding points to the feasibility of efficient, privacy-aware training strategies. Supplementary analyses of tag-specific performance are provided in Appendix B.6.

# 4 Closer Look at Memorization Dynamics During Training

Having identified high-memorization samples, we now investigate how memorization evolves throughout training. Specifically, we aim to understand *when* samples are memorized, *how* memorization persists or vanishes over time, and *what* dynamic patterns differentiate high-risk samples from others. This section formalizes memorization-related events and defines metrics to quantify memorization intensity.

## 4.1 Memorization and Forgetting: Event Definitions

We adopt the widely used relative distance ratio criterion Yoon et al. (2023); Gu et al. (2023); Fang et al. (2024) to determine whether a generated sample $\boldsymbol{x}$ memorizes a real training sample. Formally, let $\mathcal{D}$ be the training dataset and $d(\cdot,\cdot)$ be a distance metric in the input space. Let $\mathrm{NN}_1(\boldsymbol{x})$ and $\mathrm{NN}_2(\boldsymbol{x})$ denote the nearest and second-nearest training samples to $\boldsymbol{x}$, respectively. We define the distance ratio $r(\boldsymbol{x})$ as $r(\boldsymbol{x}) = \frac{d\left(\boldsymbol{x}, \mathrm{NN}_1(\boldsymbol{x}, \mathcal{D})\right)}{d\left(\boldsymbol{x}, \mathrm{NN}_2(\boldsymbol{x}, \mathcal{D})\right)}$, A sample $\boldsymbol{x}$ is considered memorized if $r(\boldsymbol{x}) < \tau$, where $\tau$ is a fixed memorization threshold (we follow prior work and use $\tau = \frac{1}{3}$).

Based on this definition, we track training-time dynamics at the level of individual real samples $\boldsymbol{x}_r \in \mathcal{D}$ by monitoring whether generated samples $\boldsymbol{x}_g$ are repeatedly flagged as memorizing $\boldsymbol{x}_r$. We define the following events: **First Memorization:** The earliest training epoch when any generated sample satisfies $r(\boldsymbol{x}_g) < \tau$ for a given $\boldsymbol{x}_r$. **Forget Event:** A transition where $\boldsymbol{x}_r$ was previously memorized but no longer satisfies the memorization criterion in subsequent epochs. **Cumulative Memorization Count:** The total number of times a sample is memorized during training. **Cumulative Forget Count:** The total number of transitions from memorization to non-memorization status.

To quantify the overall strength of memorization or forgetting, we adopt the *Memorization Area Under Curve* (Mem-AUC) Fang et al. (2024). For each real sample $\boldsymbol{x}_r$, we track its minimum distance ratios $r(\boldsymbol{x}_g)$ across generated samples over training epochs and define: Mem-AUC$(\boldsymbol{x}_r) = \int_0^1 \mathbb{P}[r(\boldsymbol{x}_g) < \tau \mid \mathrm{NN}_1(\boldsymbol{x}_g) = \boldsymbol{x}_r] \, d\tau$, where $\mathbb{P}[r(\boldsymbol{x}_g) < \tau]$ is the fraction of generated samples at each epoch that satisfy the memorization threshold $\tau$ for $\boldsymbol{x}_r$. Intuitively, higher Mem-AUC values indicate stronger and more persistent memorization behavior over time.

These event-based definitions and metrics form the basis of our dynamic analysis, where we compare high- and low-memorization samples for memorization dynamic analysis during training.

## 4.2 Temporal Dynamics of Memorization

Building on our tagging of high-memorization samples, we now investigate the temporal behavior of memorization throughout training. Figure 3 shows the cumulative proportion of samples memorized over training epochs, comparing the Top 5% most-memorized samples to the remaining Non-Top samples across three datasets.

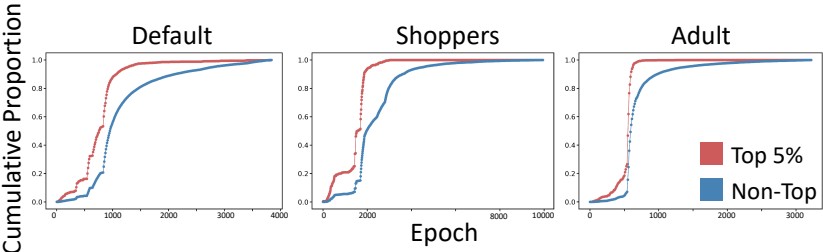

Figure 3: Cumulative proportion of samples memorized over epochs for Top 5% and Non-Top groups.

**Obs.1: Early Memorization of High-Frequency Samples.** Across all datasets—DEFAULT, SHOPPERS, and ADULT—the Top 5% samples consistently reach the memorized state earlier than the remaining data, though not excessively so. This stable temporal gap indicates that high-memorization samples are more quickly absorbed during training.

**Obs.2: Different Convergence Speeds Between Top and Non-Top Groups.** Across all three datasets, the cumulative memorization curves of the Top 5% group rise and saturate earlier than those of the Non-Top group, indicating a faster convergence toward the memorized state. In contrast, the Non-Top group exhibits a more gradual increase and reaches saturation later. This difference in convergence speed is consistent across datasets, and similar trends are also observed for the Top 10% and Top 20% subsets (Appendix B.1.2).

### 4.3   Temporal Dynamics of Forgetting

Figure 4 (A) illustrates the distribution of forget events over time, comparing Top 5% and Non-Top samples.

**Obs.1: Early Forgetting of High-Memorization Samples.** Forget events for Top samples occur earlier in training than for Non-Top samples. This pattern parallels the earlier memorization trend in Figure 3, indicating that samples memorized sooner also begin to be forgotten sooner. The alignment between early memorization and early forgetting suggests that Top samples actively participate in the learning dynamics from the very start of training. To quantify this difference, we apply the Kolmogorov–Smirnov (KS) test to compare the temporal distributions of forget events between Top and Non-Top samples. The maximum vertical separations are 8.36% (shoppers, $p \approx 0$), 4.66% (default, $p = 6.4\mathrm{e}{-}99$), and 1.28% (adult, $p = 2.5\mathrm{e}{-}11$), showing statistically significant differences in all three datasets. Moreover, when mapping the maximum separation to the time axis, the maximum horizontal gap reaches about 650 epochs for shoppers, 140 epochs for default, and 40 epochs for adult. This shows that, although the curves in Figure 4(A) appear close in value, the separation in training time is on the order of tens to hundreds of epochs, which is not small in practice.

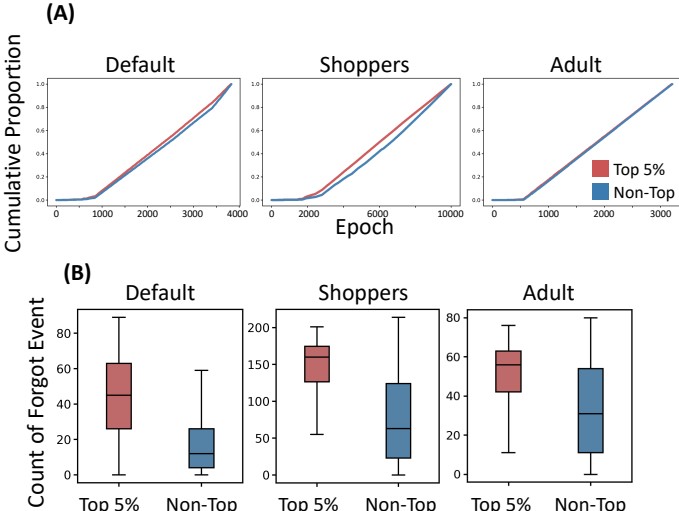

Figure 4: (A) Temporal dynamics of forget events for Top 5% and Non-Top samples. (B) Total number of forget events observed during training for Top 5% and Non-Top.

**Obs.2: Volatile Retention of Top Samples.** As shown in Figure 4(B), the Top 5% samples experience substantially more forget events throughout training. These samples frequently alternate between memorized and non-memorized states, reflecting higher volatility and sensitivity to parameter updates. Despite their early involvement in learning, they exhibit less stable retention. Consistent patterns are observed across all three datasets, with similar trends for the Top 10% and Top 20% groups detailed in Appendix B.1.3.

### 4.4   Temporal Dynamics in Memorization Intensity

To quantify how memorization strength changes during training, we track the average Mem-AUC (memorization area under curve) over epochs for Top and Non-Top groups. Mem-AUC provides a smooth measure of memorization intensity over varying thresholds.

**Obs.1: Early Spikes in Memorization Intensity for Top Samples.** As shown in Figure 5, the Top 10% samples display a pronounced spike in Mem-AUC during the early training epochs, reflecting a brief period of massive memorization intensity. In contrast, Non-Top samples exhibit a slower and more gradual increase, with overall memorization levels remaining low.

**Obs.2: Early Identification Enables Proactive Mitigation.** This early-stage peak indicates that memorization-prone samples can be detected early in training, without the need for full training-time monitoring. Such early signals enable proactive interventions—such as pruning, targeted regularization, or curriculum adjustments—to mitigate memorization risk and improve model generalization.

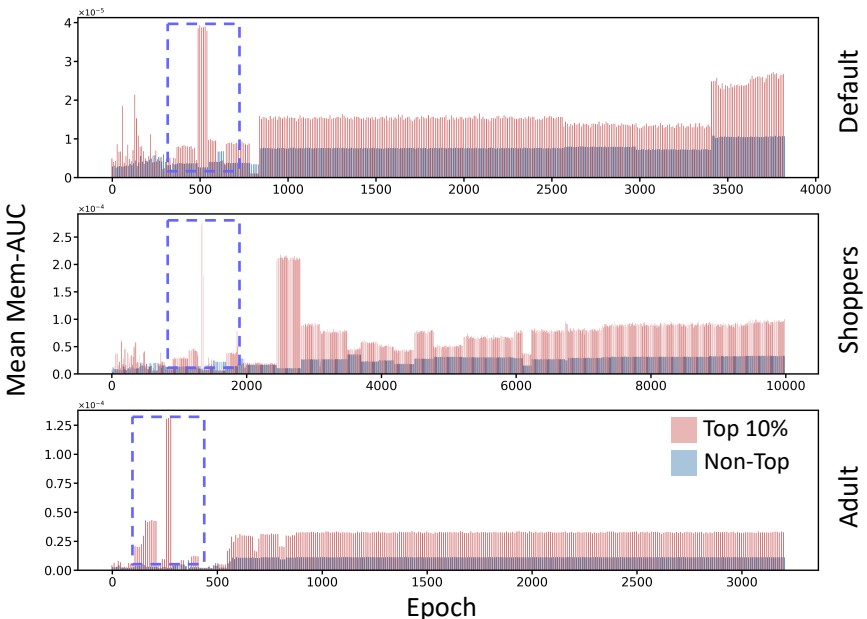

Figure 5: Mean Mem-AUC over training epochs for Top 10% and Non-Top samples. The sharp spike (**massive memorization intensity in a few epochs**) is illustrated in the blue box.

Table 2: The overview performance comparison for DynamicCut methods on different datasets. "DC" represents our proposed **DynamicCut**. "Mem. Ratio" represents memorization ratio. "Improv" represents the improvement ratio on memorization.

| | Methods | Mem. Ratio (%) ↓ | Improv. | MLE (%)↑ | $\alpha$-Precision(%)↑ | $\beta$-Recall(%)↑ | Shape Score(%)↑ | Trend Score(%)↑ | C2ST(%)↑ | DCR(%) |
|---|---|---|---|---|---|---|---|---|---|---|
| Default | TabDDPM | 19.33 ± 0.45 | - | 76.79 ± 0.69 | 98.15 ± 1.45 | 44.41 ± 0.70 | 97.58± 0.95 | 94.46 ± 0.68 | 91.85 ± 6.04 | 49.12 ± 0.94 |
| | TabDDPM+**DC** | 18.69 ± 1.04 | 3.31% ↓ | 76.42 ± 0.73 | 93.26 ± 5.44 | 42.64 ± 1.16 | 93.40 ± 3.46 | 89.65 ± 4.00 | 86.09 ± 6.48 | 50.78 ± 1.60 |
| | CTGAN | 12.83 ± 0.63 | - | 68.68 ± 0.21 | 68.95 ± 1.70 | 16.49 ± 0.55 | 85.04 ± 0.93 | 77.05 ± 2.47 | 61.89 ± 4.89 | 49.45 ± 0.80 |
| | CTGAN+**DC** | 12.60 ± 0.20 | 1.79% ↓ | 70.04 ± 0.66 | 69.06 ± 3.69 | 16.46 ± 0.97 | 85.11 ± 0.20 | 77.03 ± 0.41 | 58.03 ± 5.65 | 50.93 ± 1.82 |
| | TVAE | 17.22 ± 0.43 | - | 72.24 ± 0.36 | 82.97 ± 0.37 | 20.57 ± 0.42 | 89.37 ± 0.54 | 83.36 ± 0.99 | 52.63 ± 0.37 | 51.33 ± 0.96 |
| | TVAE+**DC** | 16.75 ± 0.60 | 2.73% ↓ | 76.87 ± 0.54 | 82.54 ± 1.97 | 21.25 ± 0.60 | 89.98 ± 0.74 | 80.36 ± 0.63 | 51.42 ± 3.67 | 50.01 ± 2.21 |
| Shoppers | TabDDPM | 31.37 ± 0.31 | - | 92.17 ± 0.32 | 93.16 ± 1.58 | 52.57 ± 1.30 | 97.08 ± 0.46 | 92.92 ± 3.27 | 86.74 ± 0.63 | 51.36 ± 0.63 |
| | TabDDPM+**DC** | 29.41 ± 2.38 | 6.66% ↓ | 91.08 ± 1.18 | 91.88 ± 2.74 | 52.74 ± 2.67 | 93.56 ± 2.89 | 89.40 ± 3.87 | 80.12 ± 4.54 | 51.87 ± 0.23 |
| | CTGAN | 19.80 ± 1.15 | - | 83.22 ± 1.30 | 85.26 ± 5.17 | 26.67 ± 1.66 | 77.73 ± 0.50 | 86.48 ± 0.69 | 67.18 ± 5.84 | 48.54 ± 0.74 |
| | CTGAN+**DC** | 19.64 ± 0.45 | 0.81% ↓ | 83.71 ± 0.44 | 89.13 ± 4.17 | 27.73 ± 0.66 | 77.78 ± 0.87 | 86.93 ± 0.34 | 69.31 ± 6.11 | 48.55 ± 1.50 |
| | TVAE | 23.17 ± 0.77 | - | 86.65 ± 0.48 | 50.32 ± 2.98 | 11.16 ± 1.33 | 74.75 ± 0.92 | 77.56 ± 1.11 | 21.29 ± 3.28 | 43.31 ± 0.65 |
| | TVAE+**DC** | 22.65 ± 0.83 | 2.24% ↓ | 86.93 ± 0.38 | 50.61 ± 1.39 | 12.49 ± 0.85 | 74.66 ± 0.73 | 79.14 ± 0.30 | 21.38 ± 2.28 | 47.61 ± 5.38 |
| Adult | TabDDPM | 31.01 ± 0.18 | - | 91.09 ± 0.07 | 93.58 ± 1.99 | 51.52 ± 2.29 | 98.84 ± 0.03 | 97.78 ± 0.07 | 94.63 ± 1.19 | 51.56 ± 0.34 |
| | TabDDPM+**DC** | 26.80 ± 1.03 | 13.58% ↓ | 90.81 ± 0.29 | 97.45 ± 1.20 | 45.54 ± 1.32 | 95.68 ± 2.55 | 93.62 ± 2.82 | 86.84 ± 3.48 | 50.42 ± 1.24 |
| | CTGAN | 21.68 ± 0.62 | - | 88.64 ± 0.32 | 78.15 ± 3.66 | 26.27 ± 0.67 | 82.43 ± 0.90 | 82.83 ± 0.93 | 63.64 ± 2.74 | 49.14 ± 0.27 |
| | CTGAN+**DC** | 16.64 ± 0.27 | 23.25% ↓ | 88.23 ± 0.67 | 74.41 ± 1.07 | 23.52 ± 0.92 | 81.08 ± 1.56 | 83.26 ± 1.00 | 61.72 ± 4.20 | 48.14 ± 0.51 |
| | TVAE | 30.78 ± 0.41 | - | 88.61 ± 0.49 | 92.34 ± 2.19 | 29.90 ± 1.00 | 82.71 ± 0.45 | 79.06 ± 0.90 | 48.67 ± 2.70 | 48.76 ± 0.25 |
| | TVAE+**DC** | 25.93 ± 1.10 | 15.76% ↓ | 88.25 ± 0.33 | 87.80 ± 0.74 | 30.57 ± 2.82 | 83.54 ± 0.56 | 78.71 ± 0.86 | 55.64 ± 2.59 | 45.96 ± 0.54 |

Table 3: The overview performance comparison for tabular models on different datasets. "DCM" represents our proposed **DynamicCutMix**.

| | Methods | Mem. Ratio (%) ↓ | Improv. | MLE (%)↑ | $\alpha$-Precision(%)↑ | $\beta$-Recall(%)↑ | Shape Score(%)↑ | Trend Score(%)↑ | C2ST(%)↑ | DCR(%) |
|---|---|---|---|---|---|---|---|---|---|---|
| Default | TabDDPM | 19.33 ± 0.45 | - | 76.79 ± 0.69 | 98.15 ± 1.45 | 44.41 ± 0.70 | 97.58± 0.95 | 94.46 ± 0.68 | 91.85 ± 6.04 | 49.12 ± 0.94 |
| | TabDDPM+**SMOTE** | 17.46 ± 0.51 | **9.66%** ↓ | 76.92 ± 0.35 | 91.19 ± 0.68 | 40.52 ± 0.65 | 94.89 ± 1.46 | 28.63 ± 2.28 | 72.73 ± 0.69 | 50.95 ± 0.38 |
| | TabDDPM+**TCM** | 16.76 ± 0.47 | **13.26%** ↓ | 76.47 ± 0.60 | 97.30 ± 0.46 | 38.72 ± 2.78 | 97.27 ± 1.74 | 93.27 ± 2.52 | 94.72 ± 3.87 | 50.23 ± 0.53 |
| | TabDDPM+**TCMP** | 18.00 ± 0.24 | **6.88%** ↓ | 76.92 ± 0.17 | 98.26 ± 0.25 | 41.92 ± 0.52 | 97.37 ± 0.09 | 91.42 ± 1.15 | 95.64 ± 0.49 | 49.75 ± 0.32 |
| | TabDDPM+**DCM** | 15.79 ± 2.06 | **18.31%** ↓ | 76.68 ± 0.76 | 91.97 ± 1.78 | 37.89 ± 0.84 | 93.62 ± 2.78 | 89.30 ± 3.02 | 87.63 ± 2.15 | 48.17 ± 2.59 |
| | CTGAN | 12.83 ± 0.63 | - | 68.68 ± 0.21 | 68.95 ± 1.70 | 16.49 ± 0.55 | 85.04 ± 0.93 | 77.05 ± 2.47 | 61.89 ± 4.89 | 49.45 ± 0.80 |
| | CTGAN+**SMOTE** | 13.56 ± 0.41 | **−5.69%** ↓ | 71.23 ± 1.54 | 65.18 ± 2.81 | 17.13 ± 0.20 | 84.31 ± 0.20 | 31.76 ± 0.11 | 56.35 ± 0.60 | 49.05 ± 2.85 |
| | CTGAN+**TCM** | 12.87 ± 0.28 | **−0.03%** ↓ | 73.21 ± 0.45 | 75.30 ± 2.12 | 16.63 ± 0.27 | 85.77 ± 0.13 | 76.69 ± 1.20 | 58.62 ± 5.34 | 49.90 ± 3.12 |
| | CTGAN+**TCMP** | 11.80 ± 0.25 | **8.03%** ↓ | 70.05 ± 0.81 | 71.33 ± 0.82 | 17.02 ± 0.60 | 85.28 ± 0.97 | 78.09 ± 0.32 | 63.67 ± 3.07 | 50.14 ± 0.60 |
| | CTGAN+**DCM** | 12.47 ± 0.07 | **2.81%** ↓ | 72.37 ± 0.39 | 71.72 ± 0.74 | 16.64 ± 0.20 | 85.78 ± 0.70 | 76.85 ± 1.62 | 57.98 ± 3.09 | 48.48 ± 0.24 |
| | TVAE | 17.22 ± 0.43 | - | 72.24 ± 0.36 | 82.97 ± 0.37 | 20.57 ± 0.42 | 89.37± 0.54 | 83.36 ± 0.99 | 52.63 ± 0.37 | 51.33 ± 0.96 |
| | TVAE+**SMOTE** | 15.64 ± 0.41 | **9.18%** ↓ | 75.41 ± 0.42 | 82.47 ± 0.42 | 20.77 ± 0.73 | 88.79 ± 0.08 | 79.34 ± 2.40 | 53.28 ± 3.26 | 49.20 ± 1.54 |
| | TVAE+**TCM** | 16.42 ± 0.37 | **4.65%** ↓ | 75.05 ± 0.66 | 80.23 ± 2.07 | 21.90 ± 2.36 | 89.65 ± 0.15 | 80.15 ± 0.75 | 52.10 ± 4.29 | 49.99 ± 1.78 |
| | TVAE+**TCMP** | 15.59 ± 0.28 | **9.47%** ↓ | 72.75 ± 0.52 | 81.57 ± 0.32 | 19.52 ± 0.39 | 89.33 ± 0.48 | 78.04 ± 6.05 | 45.60 ± 0.90 | 50.07 ± 0.52 |
| | TVAE+**DCM** | 16.32 ± 0.21 | **5.23%** ↓ | 76.27 ± 0.25 | 85.49 ± 0.41 | 21.66 ± 0.19 | 90.36 ± 0.25 | 86.76 ± 5.13 | 54.81 ± 0.79 | 51.07 ± 1.35 |
| Shoppers | TabDDPM | 31.37 ± 0.31 | - | 92.17 ± 0.32 | 93.16 ± 1.58 | 52.57 ± 1.30 | 97.08 ± 0.46 | 92.92 ± 3.27 | 86.74 ± 0.63 | 51.36 ± 0.63 |
| | TabDDPM+**SMOTE** | 26.64 ± 1.46 | **15.07%** ↓ | 89.96 ± 0.95 | 94.41 ± 4.67 | 45.22 ± 3.26 | 90.78 ± 0.49 | 83.09± 2.47 | 64.05 ± 1.44 | 51.94 ± 1.52 |
| | TabDDPM+**TCM** | 25.56 ± 1.17 | **18.51%** ↓ | 92.17 ± 0.26 | 94.41 ± 1.49 | 50.05 ± 1.59 | 97.18 ± 0.34 | 93.95± 0.51 | 86.96 ± 0.50 | 47.52± 1.81 |
| | TabDDPM+**TCMP** | 28.51 ± 0.35 | **9.12%** ↓ | 92.09 ± 0.99 | 93.43 ± 1.65 | 52.30 ± 0.73 | 97.31 ± 0.22 | 94.79± 0.30 | 87.02 ± 2.04 | 50.83 ± 0.59 |
| | TabDDPM+**DCM** | 24.24 ± 2.83 | **22.73%** ↓ | 91.46 ± 0.74 | 93.25 ± 3.87 | 46.15 ± 4.42 | 92.32 ± 5.00 | 89.11± 4.95 | 80.56 ± 4.81 | 49.74 ± 3.34 |
| | CTGAN | 19.80 ± 1.15 | - | 83.22 ± 1.30 | 85.26 ± 5.17 | 26.67 ± 1.66 | 77.73 ± 0.50 | 86.48 ± 0.69 | 67.18± 5.84 | 48.54 ± 0.74 |
| | CTGAN+**SMOTE** | 20.53 ± 1.86 | **−3.69%** ↓ | 87.09 ± 1.22 | 76.13 ± 13.12 | 26.19 ± 3.66 | 75.57 ± 2.48 | 83.94 ± 1.13 | 59.83 ± 6.67 | 50.74 ± 1.75 |
| | CTGAN+**TCM** | 19.06 ± 0.42 | **3.74%** ↓ | 84.21 ± 1.21 | 80.82 ± 7.68 | 26.50 ± 1.50 | 77.51 ± 2.15 | 85.75 ± 1.31 | 62.04 ± 6.75 | 49.90 ± 2.33 |
| | CTGAN+**TCMP** | 19.69 ± 0.35 | **0.52%** ↓ | 83.94 ± 0.80 | 82.07 ± 9.62 | 23.47 ± 0.43 | 76.15 ± 0.22 | 84.56 ± 0.40 | 64.60 ± 0.82 | 52.02 ± 0.13 |
| | CTGAN+**DCM** | 18.71 ± 0.56 | **5.51%** ↓ | 85.51 ± 0.46 | 83.51 ± 1.60 | 26.19 ± 0.82 | 78.23 ± 0.30 | 86.36 ± 0.47 | 68.85 ± 2.45 | 50.64 ± 1.11 |
| | TVAE | 23.17 ± 0.77 | - | 86.65 ± 0.48 | 50.32 ± 2.98 | 11.16 ± 1.33 | 74.75 ± 0.92 | 77.56 ± 1.11 | 21.29 ± 3.28 | 43.31 ± 0.65 |
| | TVAE+**SMOTE** | 19.77 ± 0.39 | **14.67%** ↓ | 89.71 ± 0.69 | 52.39 ± 3.94 | 13.74 ± 0.54 | 71.80 ± 0.63 | 76.20 ± 0.68 | 20.67 ± 2.34 | 47.42 ± 3.40 |
| | TVAE+**TCM** | 20.81 ± 2.38 | **10.19%** ↓ | 87.33 ± 0.90 | 49.75 ± 5.04 | 12.66 ± 3.18 | 75.47 ± 1.30 | 78.91 ± 1.30 | 21.87 ± 3.87 | 50.06 ± 1.78 |
| | TVAE+**TCMP** | 20.16 ± 0.80 | **12.98%** ↓ | 87.07 ± 0.74 | 50.69 ± 0.44 | 10.54 ± 1.68 | 74.86 ± 0.26 | 78.55± 0.48 | 21.78 ± 4.28 | 43.21 ± 0.69 |
| | TVAE+**DCM** | 20.30 ± 1.33 | **12.39%** ↓ | 88.32 ± 0.29 | 51.12 ± 2.46 | 11.98 ± 0.58 | 75.02 ± 0.97 | 79.02 ± 0.12 | 22.99 ± 2.12 | 49.98 ± 2.55 |
| Adult | TabDDPM | 31.01 ± 0.18 | - | 91.09 ± 0.07 | 93.58 ± 1.99 | 51.52 ± 2.29 | 98.84 ± 0.03 | 97.78 ± 0.07 | 94.63 ± 1.19 | 51.56 ± 0.34 |
| | TabDDPM+**SMOTE** | 28.98 ± 0.78 | **6.56%** ↓ | 90.41 ± 0.36 | 94.93 ± 1.72 | 46.10 ± 0.65 | 93.40 ± 1.12 | 90.76 ± 1.76 | 80.75 ± 0.84 | 51.82 ± 0.56 |
| | TabDDPM+**TCM** | 27.55 ± 0.19 | **11.16%** ↓ | 91.15 ± 0.06 | 94.97 ± 0.06 | 47.43 ± 1.46 | 98.65 ± 0.03 | 97.75 ± 0.07 | 85.61 ± 16.03 | 50.99 ± 0.65 |
| | TabDDPM+**TCMP** | 26.10 ± 2.11 | **15.83%** ↓ | 90.54 ± 0.17 | 92.26 ± 6.97 | 43.49 ± 3.74 | 95.10 ± 4.27 | 91.50 ± 6.53 | 84.76 ± 10.12 | 50.68 ± 0.89 |
| | TabDDPM+**DCM** | 26.27 ± 0.57 | **15.29%** ↓ | 90.81 ± 0.16 | 95.23 ± 1.14 | 45.76 ± 1.02 | 98.27 ± 0.32 | 96.71 ± 0.86 | 92.64 ± 1.65 | 50.79 ± 0.46 |
| | CTGAN | 21.68 ± 0.62 | - | 88.64 ± 0.32 | 78.15 ± 3.66 | 26.27 ± 0.67 | 82.43 ± 0.90 | 82.83 ± 0.93 | 63.64 ± 2.74 | 49.14 ± 0.27 |
| | CTGAN+**SMOTE** | 21.54 ± 0.69 | **0.65%** ↓ | 88.59 ± 0.41 | 74.96 ± 5.01 | 24.65 ± 1.44 | 82.16 ± 1.14 | 80.23 ± 0.38 | 60.01 ± 2.10 | 51.71 ± 2.15 |
| | CTGAN+**TCM** | 19.80 ± 0.48 | **8.67%** ↓ | 88.94 ± 0.64 | 76.10 ± 4.79 | 26.29 ± 1.53 | 83.26 ± 1.30 | 80.09 ± 1.89 | 61.16 ± 3.59 | 49.87 ± 0.64 |
| | CTGAN+**TCMP** | 21.20 ± 0.31 | **2.23%** ↓ | 88.63 ± 0.66 | 76.27 ± 0.60 | 25.56 ± 0.33 | 81.42 ± 0.32 | 82.11 ± 1.01 | 60.98 ± 1.67 | 50.96± 0.17 |
| | CTGAN+**DCM** | 16.53 ± 0.88 | **23.75%** ↓ | 88.45 ± 0.29 | 79.31 ± 1.04 | 23.31 ± 1.42 | 82.23 ± 0.23 | 82.80 ± 2.27 | 63.46 ± 3.43 | 48.13 ± 0.33 |
| | TVAE | 30.78 ± 0.41 | - | 88.61 ± 0.49 | 92.34 ± 2.19 | 29.90 ± 1.00 | 82.71 ± 0.45 | 79.06 ± 0.90 | 48.67 ± 2.70 | 48.76 ± 0.25 |
| | TVAE+**SMOTE** | 28.61 ± 0.16 | **7.05%** ↓ | 87.89 ± 0.32 | 89.00 ± 3.48 | 31.01 ± 2.50 | 80.65 ± 1.28 | 81.83 ± 1.64 | 52.98 ± 6.01 | 50.37 ± 0.76 |
| | TVAE+**TCM** | 29.39 ± 0.58 | **4.52%** ↓ | 87.33 ± 0.90 | 89.01 ± 5.77 | 30.87 ± 2.48 | 87.00 ± 2.13 | 82.57 ± 1.56 | 54.98 ± 7.02 | 49.43 ± 0.04 |
| | TVAE+**TCMP** | 29.88 ± 0.34 | **2.93%** ↓ | 88.42 ± 0.26 | 88.02 ± 0.72 | 29.78 ± 1.42 | 82.76 ± 0.92 | 77.92 ± 0.66 | 52.52± 7.24 | 51.35 ± 0.29 |
| | TVAE+**DCM** | 26.11 ± 0.45 | **15.17%** ↓ | 88.12 ± 0.32 | 87.54 ± 1.24 | 31.54 ± 0.38 | 83.45 ± 0.84 | 78.57 ± 0.68 | 54.85 ± 5.05 | 48.81 ± 2.10 |

---

**Algorithm 1** Pseudo-code of DynamicCut

**Require:** Training set $\mathcal{D} = \{\boldsymbol{x}_1, \ldots, \boldsymbol{x}_N\}$, early epoch number $T$, pruning ratio $p$ (default 0.1)

1: Set $k \leftarrow \lceil 0.1 \cdot T \rceil$;  ▷ Set the number of top epochs used for scoring.
2: **for** each sample $\boldsymbol{x}_i \in \mathcal{D}$ **do**
3:  Collect Mem-AUC values $\mathcal{A}_i = \{a_1(\boldsymbol{x}_i), \ldots, a_T(\boldsymbol{x}_i)\}$;  ▷ Record early memorization dynamics.
4:  Sort $\mathcal{A}_i$ in descending order;  ▷ Identify peak memorization signals.
5:  Compute memorization score $s_i \leftarrow$ mean of top-$k$ values in $\mathcal{A}_i$;  ▷ Aggregate peak memorization into a single score.
6: **end for**
7: Set threshold $\tau$ as the $(1 - p)$ quantile of $\{s_i\}_{i=1}^{N}$;  ▷ Determine the pruning cutoff.
8: Construct filtered set $\mathcal{D}_{\text{filtered}} \leftarrow \{\boldsymbol{x}_i \in \mathcal{D} \mid s_i < \tau\}$;  ▷ Remove the top-$p$ high-risk samples.
9: **return** $\mathcal{D}_{\text{filtered}}$

## 5 Methodology

### 5.1 Motivation

Our approach is driven by a series of consistent empirical findings observed in tabular diffusion models. First, memorization follows a pronounced long-tail distribution (Figure 2), with a small subset of training samples being memorized disproportionately often. These high-risk samples tend to be memorized early in training (Figure 3), exhibit volatile learning behavior with frequent forget-and-relearn cycles (Figure 4A), and contribute significantly to overall memorization. Selective removal of these samples reduces memorization much more effectively than random removal at the same rate (Table 1). Moreover, memorization intensity—measured via Mem-AUC—peaks in the early training stages (Figure 5), indicating that high-risk samples can be identified from only partial training trajectories. Together, these observations suggest that memorization is both *predictable* and *concentrated*, enabling targeted early-stage interventions to mitigate privacy risk while preserving data utility.

### 5.2 Algorithms

To operationalize these insights, we propose DynamicCut, a lightweight, model-agnostic filtering method that proactively identifies and removes memorization-prone samples based on early training dynamics. The procedure is summarized in Algorithm 1.

Based on the observation that memorization-prone samples show strong signals early in training, we propose a lightweight filtering strategy that uses early Mem-AUC dynamics. For each training sample $\boldsymbol{x}_i \in \mathcal{D} = \{\boldsymbol{x}_1, \ldots, \boldsymbol{x}_N\}$, we collect Mem-AUC values $\mathcal{A}_i = \{a_1(\boldsymbol{x}_i), \ldots, a_T(\boldsymbol{x}_i)\}$ during the first $T$ epochs. We then compute a memorization score $s_i$ as the mean of the top-$k$ highest values in $\mathcal{A}_i$, where $k = \lceil 0.1 \cdot T \rceil$. After ranking all samples by $s_i$, we remove the top $p$ ratio with the highest scores, resulting in the filtered set $\mathcal{D}_{\text{filtered}} = \{\boldsymbol{x}_i \in \mathcal{D} \mid s_i < \tau\}$, where $\tau$ is the $(1-p)$ quantile. We use $p = 0.1$ by default.

## 6 Experiments

In this section, we extensively evaluate the effectiveness of DynamicCut and DynamicCutMix across several SOTA tabular diffusion models in various datasets and compare other augmentation methods SMOTE Chawla et al. (2002), TabCutMix Fang et al. (2024) and TabCutMixPlus Fang et al. (2024).

### 6.1 Experimental Setup

**Datasets.** We utilize three real-world tabular datasets—Adult, Default, and Shoppers—each comprising both numerical and categorical features. Detailed descriptions and summary statistics for these datasets can be found in the Appendix D.4.

**Models.** We integrate DynamicCut into three generative models for tabular data: TabDDPM Kotelnikov et al. (2023), CTGAN Xu et al. (2019), and TVAE Xu et al. (2019). This integration enables a comprehensive evaluation of each model's generation quality and memorization behavior when enhanced with DynamicCut. More detailed information about these diffusion models can be found in the Appendix D.2

**Evaluation Metrics.** We evaluate the performance of synthetic data generation from two perspectives: memorization and synthetic data quality. For memorization evaluation, we generate the same number of synthetic samples as the training dataset and follow Appendix D.3 to calculate the distance between the generated and real samples. For synthetic data quality evaluation, we consider 1) low-order statistics (i.e., column-wise density and pair-wise column correlation) measured by shape score and trend score; 2) high-order metrics $\alpha$-precision and $\beta$-recall scores measuring the overall fidelity and diversity of synthetic data; 3) downstream tasks performance machine learning efficiency (MLE); 4) C2ST (Classifier Two-Sample Test) evaluates data quality by measuring how well a classifier can distinguish real from synthetic data—lower accuracy suggests better distributional alignment; 5) DCR (Distance to Closest Record) measures privacy risk by quantifying how closely a synthetic sample resembles training vs. holdout samples—lower differences

indicate better privacy preservation. The reported results are averaged over 5 independent experimental runs. More details on evaluation metrics can be found in Zhang et al. (2023); Fang et al. (2024).

Table 4: Comparison of memorization mitigation Transferability performance for tabular models under different tagging sources. "DCM" represents our proposed **DynamicCutMix**. The "Tagging" column indicates the source of memorization labels: "–" means no filtering was applied.

| | Tagging | Methods | Mem. Ratio (%) ↓ | Improv. | MLE (%)↑ | α-Precision(%)↑ | β-Recall(%)↑ | Shape Score(%)↑ | Trend Score(%)↑ | C2ST(%)↑ | DCR(%) |
|---|---|---|---|---|---|---|---|---|---|---|---|
| Default | - | CTGAN | 12.83 ± 0.63 | - | 68.68 ± 0.21 | 68.95 ± 1.70 | 16.49 ± 0.55 | 85.04 ± 0.93 | 77.05 ± 2.47 | 61.89 ± 4.89 | 49.45 ± 0.80 |
| | CTGAN | CTGAN+**DCM** | 12.47 ± 0.07 | **2.81%**↓ | 72.37 ± 0.39 | 71.72 ± 0.74 | 16.64 ± 0.20 | 85.78 ± 0.70 | 76.85 ± 1.62 | 57.98 ± 3.09 | 48.48 ± 0.24 |
| | TabDDPM | CTGAN+**DCM** | 12.69 ± 0.32 | **1.09%**↓ | 71.41± 0.89 | 68.93 ± 0.94 | 16.09 ±0.59 | 84.42 ± 1.03 | 75.38 ± 1.54 | 56.85 ± 4.31 | 50.49 ± 1.12 |
| | - | TVAE | 17.22 ± 0.43 | - | 72.24 ± 0.36 | 82.97 ± 0.37 | 20.57 ± 0.42 | 89.37± 0.54 | 83.36 ± 0.99 | 52.63 ± 0.37 | 51.33 ± 0.96 |
| | TVAE | TVAE+**DCM** | 16.32 ± 0.21 | **5.23%**↓ | 76.27 ± 0.25 | 85.49 ± 0.41 | 21.66 ± 0.19 | 90.36 ± 0.25 | 86.76 ± 5.13 | 54.81 ± 0.79 | 51.07 ± 1.35 |
| | TabDDPM | TVAE+**DCM** | 16.54 ± 0.27 | **3.95%**↓ | 73.82 ± 0.40 | 80.89 ± 1.12 | 21.47 ± 1.28 | 89.58 ± 0.19 | 78.24 ± 1.28 | 53.27 ± 4.92 | 50.92 ± 2.49 |
| Shoppers | - | CTGAN | 19.80 ± 1.15 | - | 83.22 ± 1.30 | 85.26 ± 5.17 | 26.67 ± 1.66 | 77.73 ± 0.50 | 86.48 ± 0.69 | 67.18± 5.84 | 48.54 ± 0.74 |
| | CTGAN | CTGAN+**DCM** | 18.71 ± 0.56 | **5.51%**↓ | 85.51 ± 0.46 | 83.51 ± 1.60 | 26.19 ± 0.82 | 78.23 ± 0.30 | 86.36 ± 0.47 | 68.85 ± 2.45 | 50.64 ± 1.11 |
| | TabDDPM | CTGAN+**DCM** | 18.58 ± 0.43 | **6.16%**↓ | 83.52 ± 0.53 | 80.21 ± 2.75 | 25.70 ± 1.22 | 77.49 ±1.24 | 86.40 ± 0.62 | 66.03 ± 7.67 | 50.47 ± 2.27 |
| | - | TVAE | 23.17 ± 0.77 | - | 86.65 ± 0.48 | 50.32 ± 2.98 | 11.16 ± 1.33 | 74.75 ± 0.92 | 77.56 ± 1.11 | 21.29 ± 3.28 | 43.31 ± 0.65 |
| | TVAE | TVAE+**DCM** | 20.30 ± 1.33 | **12.39%**↓ | 88.32 ± 0.29 | 51.12 ± 2.46 | 11.98 ± 0.58 | 75.02 ± 0.97 | 79.02 ± 0.12 | 22.99 ± 2.12 | 49.98 ± 2.55 |
| | TabDDPM | TVAE+**DCM** | 19.54 ± 0.95 | **14.76%**↓ | 88.43 ± 0.17 | 50.43 ± 3.45 | 12.08 ± 2.01 | 74.82 ± 0.25 | 78.37 ± 1.18 | 22.57 ± 3.68 | 47.26 ±1.81 |
| Adult | - | CTGAN | 21.68 ± 0.62 | - | 88.64 ± 0.32 | 78.15 ± 3.66 | 26.27 ± 0.67 | 82.43 ± 0.90 | 82.83 ± 0.93 | 63.64 ± 2.74 | 49.14 ± 0.27 |
| | CTGAN | CTGAN+**DCM** | 16.53 ± 0.88 | **23.75%**↓ | 88.45 ± 0.29 | 79.31 ± 1.04 | 23.31 ± 1.42 | 82.23 ± 0.23 | 82.80 ± 2.27 | 63.46 ± 3.43 | 48.13 ± 0.33 |
| | TabDDPM | CTGAN+**DCM** | 19.71 ± 0.32 | **9.09%**↓ | 87.87 ± 0.37 | 72.40 ± 4.17 | 25.56 ± 1.45 | 81.42 ± 1.91 | 81.15 ± 1.23 | 58.84 ± 2.46 | 50.00 ±0.19 |
| | - | TVAE | 30.78 ± 0.41 | - | 88.61 ± 0.49 | 92.34 ± 2.19 | 29.90 ± 1.00 | 82.71 ± 0.45 | 79.06 ± 0.90 | 48.67 ± 2.70 | 48.76 ± 0.25 |
| | TVAE | TVAE+**DCM** | 26.11 ± 0.45 | **15.17%**↓ | 88.12 ± 0.32 | 87.54 ± 1.24 | 31.54 ± 0.38 | 83.45 ± 0.84 | 78.57 ± 0.68 | 54.85 ± 5.05 | 48.81 ± 2.10 |
| | TabDDPM | TVAE+**DCM** | 28.68 ± 0.21 | **6.82%**↓ | 89.23 ± 0.38 | 85.50 ± 4.09 | 30.23 ± 3.81 | 82.36 ± 1.96 | 78.80 ± 2.50 | 54.48 ± 8.01 | 49.57 ± 0.35 |

## 6.2 Unified Evaluation of Memorization under Data Quality and Training Dynamics

Table 2 presents a comprehensive comparison of generative models with and without our proposed DynamicCut strategy across three datasets: DEFAULT, SHOPPERS, and ADULT.

**Obs.1: Consistent Memorization Reduction with DynamicCut.** Across all model–dataset combinations, applying DynamicCut consistently lowers the memorization ratio. For example, TabDDPM's memorization ratio on the ADULT dataset decreases from 31.01% to 26.80%, corresponding to a 13.58% relative reduction. Similar decreases are observed on DEFAULT and SHOPPERS (3.31% and 6.66%, respectively). Furthermore, other model families such as CTGAN and TVAE also benefit from DynamicCut, achieving reductions of up to 23.25% and 15.76% on ADULT. These consistent improvements demonstrate that DynamicCut effectively mitigates memorization across diverse architectures and data regimes.

**Obs.2: Preservation of Downstream Utility.** The decrease in memorization does not entail a major loss of downstream performance. Although α-Precision and β-Recall occasionally show minor declines, other metrics such as Shape Score, Trend Score, and MLE remain stable or even improve. For instance, on DEFAULT, TabDDPM+DynamicCut maintains high fidelity in both shape (93.40%) and trend (89.65%) scores, closely matching the baseline while achieving better memorization control.

These results highlight that DynamicCut offers a principled and effective solution to control memorization risk without significantly sacrificing generation quality, making it a practical add-on for improving both the fidelity and safety of tabular generative models. Additional tag ratio and hyperparameter studies are provided in Appendix B.1 and B.4.

## 6.3 DynamicCut with TabCutMix (DynamicCutMix): Combined Effect and Evaluation

Table 3 summarizes the performance of our proposed DCM method, which combines memorization-aware filtering (DC) with data augmentation (TCM). We evaluate DCM on three generative models (TabDDPM, CTGAN, TVAE) and three datasets (DEFAULT, SHOPPERS, ADULT), comparing it to strong baselines including TCM, TCMP, and SMOTE.

**Obs.1: Superior Memorization Reduction by DCM.** Across all model–dataset combinations, DCM consistently achieves the lowest memorization ratios. For instance, on DEFAULT with TabDDPM, DCM reduces memorization to 15.79%, outperforming both TCM (16.76%) and TCMP (18.00%). Similar trends are observed for other models and datasets, confirming DCM's effectiveness in suppressing memorization.

**Obs.2: Strong Utility Preservation of DCM.** In addition to reducing memorization, DCM maintains high downstream utility. On ADULT, TabDDPM+DCM attains a Shape Score of 98.27% and a Trend Score of 96.71%, surpassing SMOTE and matching or exceeding TCM and TCMP. These results highlight DCM's ability to achieve a favorable balance between memorization mitigation and sample fidelity.

Table 5: The real and generative samples by TabDDPM and TabDDPM with DCM in Adult dataset. DCM represents DynamicCutMix.

| Samples | Age | Workclass | fnlwgt | Education | Education.num | Marital Status | Occupation | Relationship | Race | Sex | Capital Gain | Capital Loss | Hours per Week | Native Country | Income |
|---|---|---|---|---|---|---|---|---|---|---|---|---|---|---|---|
| Real | 42.0 | Private | 108502.0 | HS-grad | 9.0 | Divorced | Sales | Not-in-family | White | Female | 0.0 | 0.0 | 42.0 | United-States | <=50K |
| TabDDPM | 55.0 | Private | 107841.53 | HS-grad | 9.0 | Divorced | Sales | Not-in-family | White | Female | 0.0 | 0.0 | 68.07 | United-States | <=50K |
| TabDDPM+**DCM** | 38.0 | Private | 110753.5 | HS-grad | 9.0 | Divorced | Sales | Not-in-family | Black | Female | 0.0 | 0.0 | 40.0 | United-States | <=50K |
| Real | 21.0 | Private | 185251.0 | HS-grad | 9.0 | Never-married | Sales | Own-child | White | Male | 0.0 | 0.0 | 40.0 | United-States | <=50K |
| TabDDPM | 24.54 | Private | 185105.25 | HS-grad | 9.0 | Never-married | Sales | Own-child | White | Male | 0.0 | 0.0 | 24.78 | United-States | <=50K |
| TabDDPM+**DCM** | 18.0 | Private | 187309.42 | HS-grad | 9.0 | Never-married | Sales | Own-child | White | Male | 0.0 | 0.0 | 25.0 | United-States | <=50K |

## 6.4 Cross-Model Transferability of Memorization Signals

Table 4 evaluates whether memorization-prone samples identified by TabDDPM generalize to other models. We compare standard DCM, which uses model-specific memorization signals (e.g., CTGAN+DCM), against **DCM-TabDDPM**, which uses labels derived from TabDDPM's training and applies them to CTGAN and TVAE.

**Obs.1: Cross-model Transferability of DCM Labels.** TabDDPM-derived labels exhibit strong transferability across model families. For example, on ADULT, CTGAN+DCM reduces memorization by 23.75%, while CTGAN+DCM-TabDDPM still achieves a 9.09% reduction. Similar patterns hold for TVAE across datasets. Although slightly less effective than model-specific labels, DCM-TabDDPM continues to deliver consistent memorization reduction benefits.

**Obs.2: Retained Utility under Cross-model Application.** DCM-TabDDPM also preserves model utility. Core metrics such as Shape Score and Trend Score remain largely stable, with only minor performance trade-offs. For instance, on DEFAULT, TVAE+DCM-TabDDPM attains a Shape Score of 89.58%, closely matching the 90.36% achieved by the model-specific DCM. These results confirm that label transfer across models can mitigate memorization without compromising data quality.

These results suggest that memorization signals from TabDDPM are robust and broadly useful, making DCM applicable even in settings without access to full training traces from the target model. Additional experimental results are provided in Appendix B.5.

## 6.5 Case Study on Adult Dataset: Real vs. Generated Samples

Table 5 presents representative examples from the Adult dataset comparing real samples, TabDDPM-generated samples, and those generated with DCM.

**Obs.1: Evidence of Sample-level Memorization in TabDDPM.** TabDDPM often produces samples that closely resemble real records, with categorical attributes largely unchanged and only minor fluctuations in numerical fields. This high structural similarity indicates direct sample-level memorization.

**Obs.2: Enhanced Diversity with DCM.** DCM introduces greater variability in continuous attributes such as age and working hours while preserving coherence in categorical features. For example, the distance ratio rises from 0.0971 to 0.7598, suggesting that the generated sample diverges from the original and is no longer memorized.

## 7 Conclusions

We study memorization dynamics in tabular diffusion models from a data-centric perspective and find that a small subset of samples is memorized early and disproportionately. Leveraging this, we propose DynamicCut to filter high-risk samples using early memorization signals, and DynamicCutMix to combine filtering with feature-level augmentation. Our methods are simple, model-agnostic, and effective across datasets and architectures, reducing memorization while preserving data quality and generalizing across generative models. Limitations and future directions are discussed in Appendix C.1 and C.2.

**Acknowledgments**

This work made use of the High Performance Computing Resource in the Core Facility for Advanced Research Computing at Case Western Reserve University. This work was supported in part by NSF CCF-2200255, NSF CCF-2006780, NSF IIS-2027667, NIH U01AG073323, NIH R01HG009658, NIH 1R01HL159170-01A1, as well as by the Clinical and Translational Science Collaborative of Cleveland, which is funded by the National Institutes of Health, National Center for Advancing Translational Sciences, through the Clinical and Translational Science Award UL1TR002548.

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

# A  Appendix

# B  More Experimental Results

## B.1  Experiments on More Tag Ratios

### B.1.1  Overall Performance Comparison

To understand how the tag ratio $p$ used to identify high-memorization samples affects DynamicCut's effectiveness, we conduct experiments across five different tag ratios: $p = 1\%, 3\%, 5\%, 10\%$, and $20\%$ on the SHOPPERS and ADULT datasets, as shown in Table 6. We note that the distribution of memorization frequency in 2 is typically long-tailed, and only a small fraction of samples are replicated many times. Importantly, DynamicCut does not rely on a hard frequency cutoff, but instead ranks samples by their warm-up Mem-AUC scores and selects the top-$p\%$ most memorization-prone samples. We treat $p$ as an empirical design choice rather than a theoretically optimal constant, and study its effect below. We have the following observations:

**Obs.1: Optimal Filtering Threshold at $p = 10\%$.** Across both SHOPPERS and ADULT, setting $p = 10\%$ consistently achieves the largest reduction in memorization—22.73% and 15.29%, respectively—outperforming smaller thresholds ($p = 1\%, 3\%, 5\%$) and the larger threshold ($p = 20\%$). This suggests that a moderate filtering ratio strikes a balance between aggressiveness and reliability in identifying memorization-prone samples.

**Obs.2: Trade-offs at Extreme Thresholds.** A lower threshold ($p = 1\%, 3\%, 5\%$) yields weaker memorization reduction and, in some cases, greater degradation in utility metrics such as $\beta$-Recall. Conversely, a higher threshold ($p = 20\%$) leads to less precise filtering, resulting in weaker memorization mitigation and inconsistent improvements in utility performance.

Overall, these results suggest that the threshold $p$ should be chosen in a moderate range rather than being overly conservative or overly aggressive. In our experiments, using a 10% threshold for filtering provides the best trade-off between memorization suppression and generation quality, and we therefore adopt $p = 10\%$ as the default setting for DynamicCut.

Table 6: Ablation study on the ratio $p$ used for selecting high-memorization samples to filter in DynamicCut. "DCM 1%", "DCM 3%", "DCM 5%", "DCM 10%", and "DCM 20%" refer to retaining only the top-$p$ fraction of memorized samples (according to warm-up Mem-AUC) for filtering. "Improv." reports the relative memorization reduction compared to base TabDDPM. Results are shown for SHOPPERS and ADULT.

| | Methods | Mem. Ratio (%) ↓ | Improv. | MLE (%)↑ | $\alpha$-Precision(%)↑ | $\beta$-Recall(%)↑ | Shape Score(%)↑ | Trend Score(%)↑ | C2ST(%)↑ | DCR(%) |
|---|---|---|---|---|---|---|---|---|---|---|
| Shoppers | TabDDPM | $31.37 \pm 0.31$ | - | $92.17 \pm 0.32$ | $93.16 \pm 1.58$ | $52.57 \pm 1.30$ | $97.08 \pm 0.46$ | $92.92 \pm 3.27$ | $86.74 \pm 0.63$ | $51.36 \pm 0.63$ |
| | TabDDPM+**DCM 1%** | $29.44 \pm 0.83$ | **6.16%** ↓ | $92.25 \pm 0.50$ | $94.18 \pm 1.50$ | $51.76 \pm 1.17$ | $95.81 \pm 0.54$ | $92.71 \pm 1.52$ | $84.28 \pm 3.54$ | $50.56 \pm 2.03$ |
| | TabDDPM+**DCM 3%** | $29.46 \pm 0.95$ | **6.09%** ↓ | $91.73 \pm 0.63$ | $93.59 \pm 2.63$ | $52.20 \pm 0.54$ | $96.78 \pm 0.52$ | $92.69 \pm 1.49$ | $85.81 \pm 0.85$ | $51.09 \pm 1.05$ |
| | TabDDPM+**DCM 5%** | $28.43 \pm 1.76$ | **9.37%** ↓ | $93.30 \pm 0.22$ | $94.49 \pm 1.87$ | $49.07 \pm 2.35$ | $95.98 \pm 1.14$ | $92.04 \pm 1.81$ | $84.66 \pm 3.35$ | $51.28 \pm 1.28$ |
| | TabDDPM+**DCM 10%** | $24.24 \pm 2.83$ | **22.73%** ↓ | $91.46 \pm 0.74$ | $93.25 \pm 3.87$ | $46.15 \pm 4.42$ | $92.32 \pm 5.00$ | $89.11 \pm 4.95$ | $80.56 \pm 4.81$ | $49.74 \pm 3.34$ |
| | TabDDPM+**DCM 20%** | $29.40 \pm 0.40$ | **6.28%** ↓ | $91.58 \pm 0.51$ | $91.66 \pm 2.77$ | $52.98 \pm 1.99$ | $95.30 \pm 1.88$ | $91.83 \pm 1.35$ | $82.69 \pm 4.29$ | $52.44 \pm 0.33$ |
| Adult | TabDDPM | $31.01 \pm 0.18$ | - | $91.09 \pm 0.07$ | $93.58 \pm 1.99$ | $51.52 \pm 2.29$ | $98.84 \pm 0.03$ | $97.78 \pm 0.07$ | $94.63 \pm 1.19$ | $51.56 \pm 0.34$ |
| | TabDDPM+**DCM 1%** | $28.26 \pm 1.71$ | **9.78%** ↓ | $90.81 \pm 0.24$ | $95.59 \pm 0.98$ | $46.18 \pm 2.76$ | $97.89 \pm 1.39$ | $96.05 \pm 2.59$ | $90.97 \pm 3.58$ | $50.44 \pm 1.03$ |
| | TabDDPM+**DCM 3%** | $27.75 \pm 0.72$ | **10.51%** ↓ | $90.46 \pm 0.14$ | $95.37 \pm 0.83$ | $44.46 \pm 2.11$ | $97.61 \pm 1.86$ | $94.98 \pm 3.94$ | $90.53 \pm 3.27$ | $50.61 \pm 1.06$ |
| | TabDDPM+**DCM 5%** | $27.63 \pm 1.41$ | **10.90%** ↓ | $90.91 \pm 0.26$ | $95.73 \pm 2.92$ | $45.56 \pm 2.63$ | $98.08 \pm 0.48$ | $96.14 \pm 1.50$ | $90.21 \pm 3.35$ | $51.06 \pm 0.56$ |
| | TabDDPM+**DCM 10%** | $26.27 \pm 0.57$ | **15.29%** ↓ | $90.81 \pm 0.16$ | $95.23 \pm 1.14$ | $45.76 \pm 1.02$ | $98.27 \pm 0.32$ | $96.71 \pm 0.86$ | $92.64 \pm 1.65$ | $50.79 \pm 0.46$ |
| | TabDDPM+**DCM 20%** | $26.09 \pm 0.82$ | **15.87%** ↓ | $90.63 \pm 0.31$ | $94.75 \pm 2.10$ | $44.45 \pm 0.83$ | $98.03 \pm 0.50$ | $94.79 \pm 1.88$ | $91.01 \pm 2.39$ | $50.85 \pm 0.24$ |

### B.1.2 Temporal Dynamics of Memorization

Figure 6 presents the cumulative proportion of memorized samples across training epochs for Top and Non-Top groups. Compared to the 5% results shown in the main paper, this figure includes additional curves for Top 10% and Top 20% samples, demonstrating that the early memorization trend persists across broader thresholds.

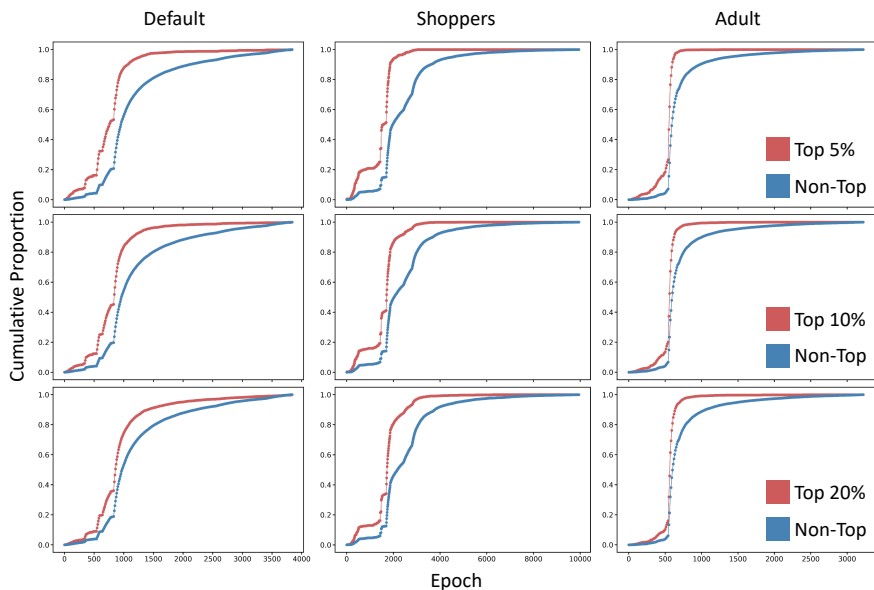

Figure 6: Cumulative proportion of samples memorized over epochs for Top and Non-Top groups.

### B.1.3 Temporal Dynamics of Forgetting

Figure 7 and Figure 8 extend the forgetting analysis in the main paper by including Top 10% and Top 20% groups in addition to the Top 5%. The results confirm that higher-memorization samples consistently exhibit earlier and more frequent forgetting behavior across all thresholds.

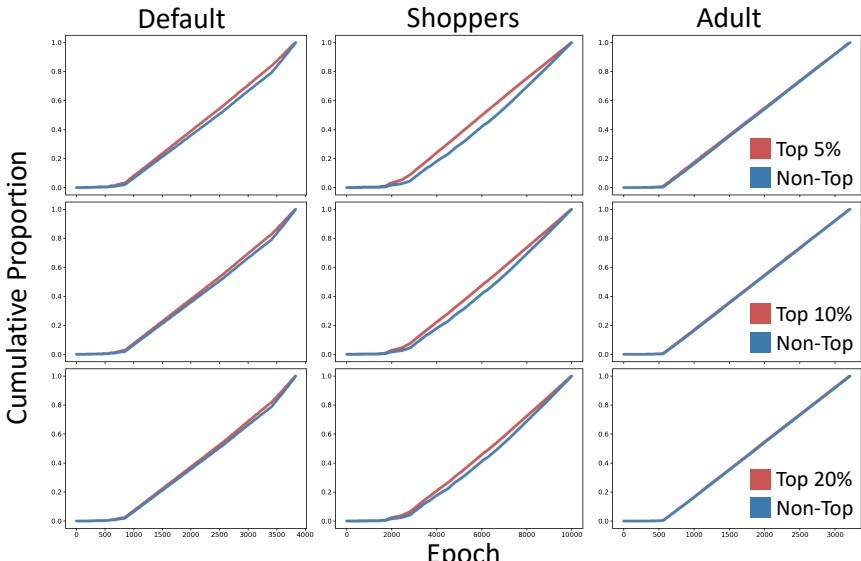

Figure 7: Dynamic changes in the number of forget events over training epochs for different groups.

### B.2 Experiments on Different Warm-up Window Lengths

To understand how the choice of the warm-up window length affects DynamicCut's effectiveness, we conduct experiments with three different window settings on each dataset: 600, 700, and 800 epochs for DEFAULT, and 2000, 2100, and 2200 epochs for SHOPPERS, as shown in Table 7. We emphasize that the warm-up window is a data-dependent design choice rather than a fixed hyperparameter, since different datasets can exhibit different training dynamics. Accordingly, our goal here is not to identify a single universally optimal window, but to examine how reasonable variations of this choice influence both memorization mitigation and generation quality. We have the following observations:

**Obs.1: The Optimal Window Length Is Dataset-Dependent.** We observe that the best warm-up window length is not identical across datasets, indicating that this choice is inherently dataset-dependent. While the default window (600 epochs for DEFAULT and 2000 epochs for SHOPPERS) achieves strong memorization reduction, extending the window to 700/800 or 2100/2200 epochs leads to different degrees of performance change across the two datasets. This further confirms that the warm-up window should be viewed as a data-dependent design choice rather than a fixed hyperparameter.

**Obs.2: Overly Long Windows Degrade Memorization Mitigation.** When the warm-up window is extended to 700 or 800 epochs on DEFAULT, and to 2100 or 2200 epochs on SHOPPERS, we consistently observe weaker memorization reduction and, in some cases, slight degradation in utility metrics. This suggests that using an overly long window can dilute the early memorization signal and reduce the precision of identifying high-risk samples, leading to less effective filtering.

These results indicate that the warm-up window should focus on early training dynamics and be adjusted based on the dataset, rather than being set to an excessively large value. In practice, moderate window lengths provide a better balance between stable memorization suppression and generation quality, and DynamicCut remains robust under reasonable variations of this design choice.

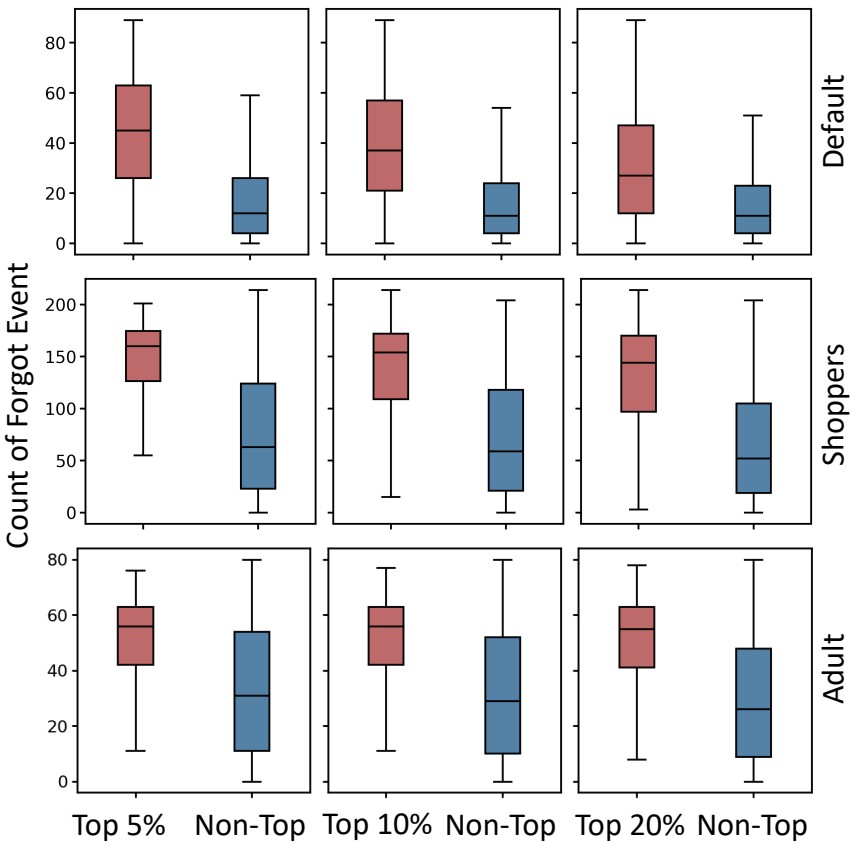

Figure 8: Dynamic changes in the number of forget events over training epochs for different groups.

Table 7: Ablation study on different warm-up window lengths used in DynamicCut. "Improv." reports the relative memorization reduction compared to base TabDDPM.

| Dataset | Window | Mem. Ratio (%) ↓ | Improv. | MLE (%)↑ | $\alpha$-Precision(%)↑ | $\beta$-Recall(%)↑ | Shape Score(%)↑ | Trend Score(%)↑ | C2ST(%)↑ | DCR(%) |
|---|---|---|---|---|---|---|---|---|---|---|
| | 600 | $15.79 \pm 2.06$ | **18.31%** ↓ | $76.68 \pm 0.76$ | $91.97 \pm 1.78$ | $37.89 \pm 0.84$ | $93.62 \pm 2.78$ | $89.30 \pm 3.02$ | $87.63 \pm 2.15$ | $48.17 \pm 2.59$ |
| Default | 700 | $18.14 \pm 2.36$ | **6.17%** ↓ | $78.06 \pm 1.33$ | $92.36 \pm 4.21$ | $42.19 \pm 1.13$ | $93.21 \pm 1.44$ | $90.25 \pm 1.72$ | $84.66 \pm 3.41$ | $49.47 \pm 1.42$ |
| | 800 | $19.10 \pm 0.46$ | **1.21%** ↓ | $77.75 \pm 0.53$ | $90.14 \pm 6.30$ | $43.28 \pm 5.31$ | $93.66 \pm 7.08$ | $91.10 \pm 5.83$ | $85.78 \pm 4.28$ | $51.10 \pm 0.67$ |
| | 2000 | $24.24 \pm 2.83$ | **22.73%** ↓ | $91.46 \pm 0.74$ | $93.25 \pm 3.87$ | $46.15 \pm 4.42$ | $92.32 \pm 5.00$ | $89.11 \pm 4.95$ | $80.56 \pm 4.81$ | $49.74 \pm 3.34$ |
| Shoppers | 2100 | $28.11 \pm 2.00$ | **10.39%** ↓ | $90.31 \pm 1.01$ | $94.08 \pm 1.08$ | $51.07 \pm 3.17$ | $95.03 \pm 1.66$ | $91.19 \pm 2.41$ | $81.59 \pm 5.22$ | $51.00 \pm 0.25$ |
| | 2200 | $29.69 \pm 0.56$ | **5.36%** ↓ | $91.30 \pm 0.27$ | $93.04 \pm 1.45$ | $53.25 \pm 1.07$ | $94.60 \pm 2.33$ | $91.89 \pm 0.73$ | $81.76 \pm 3.96$ | $51.66 \pm 0.57$ |

## B.3 Experiments on More Datasets

To further assess the robustness and transferability of DCM, we conduct additional experiments on two diverse datasets, as shown in Table 8: CARDIO and WILT. These datasets differ significantly from the main benchmarks in size, structure, and difficulty, offering a more comprehensive evaluation of DCM's generalizability. We compare TabDDPM enhanced with DCM against several strong augmentation baselines including SMOTE, TCM, and TCMP.

**Obs: Balanced Performance of DCM on Complex Datasets.** DCM maintains competitive or superior performance across both datasets. On CARDIO, although TCMP achieves the lowest memorization ratio (23.05%), DCM attains a comparably low value (24.07%) while delivering the highest Shape Score (99.19%), a top-level Trend Score (98.72%), and the best DCR (50.93%). This demonstrates DCM's strong ability to balance diversity and fidelity in sample generation.

Table 8: Additional evaluation of tabular generative models on supplementary datasets. "DCM" denotes our proposed **DynamicCutMix** method.

| | Methods | Mem. Ratio (%) ↓ | Improv. | MLE (%)↑ | α-Precision(%)↑ | β-Recall(%)↑ | Shape Score(%)↑ | Trend Score(%)↑ | C2ST(%)↑ | DCR(%) |
|---|---|---|---|---|---|---|---|---|---|---|
| | TabDDPM | 24.63 ± 0.18 | - | 80.24 ± 0.78 | 99.14 ± 0.15 | 49.11 ± 0.17 | 99.61 ± 0.03 | 98.95 ± 0.30 | 99.43 ± 0.55 | 49.91 ± 0.36 |
| | TabDDPM+**SMOTE** | 23.10 ± 0.69 | **6.21%** ↓ | 79.47 ± 0.45 | 96.35 ± 2.38 | 47.44 ± 1.17 | 96.58 ± 0.75 | 94.39 ± 1.28 | 85.43 ± 1.69 | 50.73 ± 0.19 |
| Cardio | TabDDPM+**TCM** | 23.05 ± 0.38 | **6.40%** ↓ | 79.71 ± 0.58 | 97.82 ± 2.05 | 48.37 ± 1.11 | 98.66 ± 1.35 | 95.86 ± 3.43 | 96.31 ± 0.42 | 49.24 ± 1.58 |
| | TabDDPM+**TCMP** | 23.54 ± 0.34 | **4.43%** ↓ | 79.82 ± 0.27 | 98.71 ± 0.49 | 48.87 ± 0.34 | 98.88± 0.62 | 98.67 ± 0.20 | 96.31 ± 0.42 | 49.34 ± 0.38 |
| | TabDDPM+**DCM** | 24.07 ± 0.24 | **2.27%** ↓ | 79.41 ± 0.46 | 98.92 ± 0.62 | 48.84 ± 0.14 | 99.19 ± 0.36 | 98.72 ± 0.32 | 97.38 ± 0.65 | 50.93 ± 0.40 |
| | TabDDPM | 98.48 ± 0.35 | - | 99.32 ± 0.58 | 98.63 ± 0.73 | 50.53 ± 0.47 | 98.58 ± 1.51 | 98.48 ± 0.35 | 98.63 ± 1.68 | 52.47 ± 0.54 |
| | TabDDPM+**SMOTE** | 96.78 ± 0.41 | **1.72%** ↓ | 99.22 ± 0.54 | 79.49 ± 0.78 | 43.76 ± 1.37 | 91.09 ± 0.50 | 88.04 ± 4.73 | 76.66 ± 2.61 | 50.29 ± 0.28 |
| Wilt | TabDDPM+**TCM** | 97.17 ± 0.12 | **1.33%** ↓ | 99.22 ± 0.38 | 97.93 ± 1.01 | 48.47 ± 1.17 | 97.31 ± 1.28 | 95.71 ± 2.49 | 96.92 ± 1.72 | 48.75 ± 2.18 |
| | TabDDPM+**TCMP** | 96.75 ± 0.67 | **1.76%** ↓ | 99.52 ± 0.37 | 98.55 ± 0.14 | 49.36 ± 0.99 | 97.12 ± 0.84 | 96.81 ± 0.63 | 96.76 ± 3.44 | 45.52 ± 1.35 |
| | TabDDPM+**DCM** | 96.59 ± 0.32 | **1.92%** ↓ | 98.49 ± 0.42 | 98.84 ± 0.25 | 47.00 ± 0.40 | 97.02 ± 0.88 | 95.75 ± 2.18 | 93.62 ± 0.67 | 51.08 ± 0.53 |

On WILT, DCM outperforms all other methods in key distributional metrics. While reducing the memory ratio to 96.59%, DCM achieves the highest $\alpha$-Precision (98.84%), a strong Trend Score (95.75%), and a superior DCR (51.08%), surpassing both TCMP and SMOTE. Notably, DCM avoids the significant performance degradation in $\beta$-Recall and C2ST observed with SMOTE, highlighting its ability to preserve minority modes without compromising sample quality.

These results provide further evidence that DCM generalizes beyond standard benchmarks and remains effective across a range of dataset characteristics.

## B.4 Hyperparameter Study on Pooling Strategy

We examined how different pooling strategies for warm-up Mem-AUC scores influence DynamicCut's ability to spot high-memorization samples. Three variants were compared: (1) **Mean pooling** (DCM-Mean) – averages Mem-AUC across all warm-up epochs; (2) **Max pooling** (DCM-Max) – takes each sample's single highest Mem-AUC., and (3) **top-10% mean** pooling (DCM-Top-Mean) – averages only the top-10% highest Mem-AUC epochs for each sample. As shown in Table 9, we have the following observations:

**Obs.1: Top-10% Mean Achieves the Most Effective Memorization Mitigation.** The Top-10% mean consistently yields the greatest reduction in memorization across both datasets, decreasing memorization by 18.31% on DEFAULT and 22.73% on SHOPPERS relative to the TabDDPM baseline. This confirms that targeted averaging among high-memorization samples best balances precision and diversity in filtering.

**Obs.2: Max Pooling Trades Memorization Control for Slight Utility Gains.** DCM-Max occasionally achieves marginal improvements in utility metrics such as MLE, but its memorization reduction is smaller and less consistent. This indicates that aggressive selection may favor utility preservation at the cost of weaker memorization suppression.

**Obs.3: Mean Pooling Provides a Balanced but Suboptimal Trade-off.** DCM-Mean delivers moderate improvements across both memorization and utility metrics but fails to outperform the Top-10% mean in either privacy protection or sample quality. These results highlight that simple averaging offers stability but lacks the targeted precision needed for optimal trade-offs.

These results suggest that computing the memorization score based on the most salient epochs offers a more reliable signal for identifying high-risk samples. Therefore, we adopt **top-10% mean** as the default pooling method in our DynamicCut implementation.

## B.5 Experiments on Transferability

To further evaluate the transferability of memorization labels across generative models, we visualize memorization ratios under different labeling-training combinations for each dataset in Figure 9. For each heatmap, the row indicates the model used to generate memorization labels (e.g., TabDDPM), and the column indicates the model trained using those labels for DCM filtering.

Table 9: Comparison of different pooling strategies used to compute memorization scores for filtering. "DCM-Mean," "DCM-Max," and "DCM-Top-Mean" denote the use of mean, max, and top-10% mean pooling over the warm-up phase Mem-AUC sequence, respectively. "Improv." indicates the relative memorization reduction over the base TabDDPM.

| | Methods | Mem. Ratio (%) ↓ | Improv. | MLE (%)↑ | α-Precision(%)↑ | β-Recall(%)↑ | Shape Score(%)↑ | Trend Score(%)↑ | C2ST(%)↑ | DCR(%) |
|---|---|---|---|---|---|---|---|---|---|---|
| Default | TabDDPM | 19.33 ± 0.45 | - | 76.79 ± 0.69 | 98.15 ± 1.45 | 44.41 ± 0.70 | 97.58± 0.95 | 94.46 ± 0.68 | 91.85 ± 6.04 | 49.12 ± 0.94 |
| | TabDDPM+**DCM-Mean** | 17.12 ± 0.78 | **11.43%** ↓ | 79.55 ± 0.70 | 95.92 ± 3.21 | 38.94 ± 0.98 | 96.21 ± 1.33 | 90.23 ± 0.88 | 87.35 ± 4.42 | 50.64 ± 0.94 |
| | TabDDPM+**DCM-Max** | 16.67 ± 1.22 | **13.76%** ↓ | 80.09 ± 0.55 | 91.08 ± 7.60 | 37.93 ± 3.26 | 94.59 ± 3.32 | 90.51 ± 3.52 | 88.47 ± 7.36 | 49.32 ± 2.63 |
| | TabDDPM+**DCM-Top-Mean** | 15.79 ± 2.06 | **18.31%** ↓ | 76.68 ± 0.76 | 91.97 ± 1.78 | 37.89 ± 0.84 | 93.62 ± 2.78 | 89.30 ± 3.02 | 87.63 ± 2.15 | 48.17 ± 2.59 |
| Shoppers | TabDDPM | 31.37 ± 0.31 | - | 92.17 ± 0.32 | 93.16 ± 1.58 | 52.57 ± 1.30 | 97.08 ± 0.46 | 92.92 ± 3.27 | 86.74 ± 0.63 | 51.36 ± 0.63 |
| | TabDDPM+**DCM-Mean** | 27.33 ± 1.12 | **12.88%** ↓ | 92.58 ± 0.42 | 95.66 ± 0.61 | 48.54 ± 0.70 | 94.12 ± 2.01 | 91.47 ± 1.07 | 82.01 ± 2.73 | 51.00 ± 2.91 |
| | TabDDPM+**DCM-Max** | 26.68 ± 2.75 | **14.95%** ↓ | 93.21 ± 0.32 | 91.93 ± 6.24 | 47.97 ± 3.22 | 94.34 ± 4.12 | 90.50 ± 5.55 | 80.66 ± 11.39 | 49.59 ± 1.86 |
| | TabDDPM+**DCM-Top-Mean** | 24.24 ± 2.83 | **22.73%** ↓ | 91.46 ± 0.74 | 93.25 ± 3.87 | 46.15 ± 4.42 | 92.32 ± 5.00 | 89.11± 4.95 | 80.56 ± 4.81 | 49.74 ± 3.34 |

Table 10: AUC scores of memorization tag classifiers trained using warm-up phase labels across three datasets. Higher AUC indicates better agreement between early-phase tags and final high memorization frequency samples.

| Dataset | Average AUC |
|---|---|
| Default | 73.78% |
| Shoppers | 80.14% |
| Adult | 76.52% |

**Obs.1: Model-specific Labels Provide the Most Precise Filtering.** Across all three datasets—DEFAULT, SHOPPERS, and ADULT—model-specific labels (diagonal entries) consistently yield the lowest memorization ratios. This indicates that labels generated by the same model offer the most accurate guidance for identifying and filtering memorization-prone samples during training.

**Obs.2: TabDDPM Labels Generalize Effectively Across Models.** Labels derived from TabDDPM transfer well to other model architectures, achieving the second-best memorization reduction in most off-diagonal settings. For instance, TabDDPM-generated labels substantially reduce memorization when applied to CTGAN and TVAE, outperforming labels produced by weaker models such as TVAE. These results suggest that TabDDPM captures structural regularities that are broadly useful for cross-model memorization mitigation.

These results confirm that memorization-prone samples identified by a strong base model (e.g., TabDDPM) can be effectively reused across other generative models for filtering, demonstrating the robustness and practical value of our transferability approach.

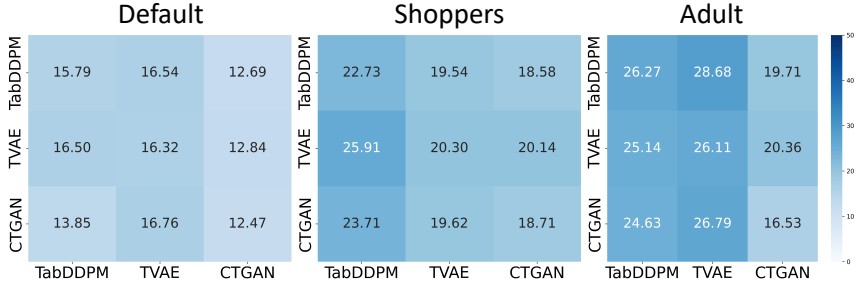

Figure 9: Heatmaps showing memorization ratios (%) for different label-source and model-target combinations on DEFAULT, SHOPPERS, and ADULT datasets. Rows denote the model used to generate memorization labels, and columns denote the model trained with those labels.

### B.6 Experiments on Memorization Tag Performance

To assess the quality of memorization labels generated during the warm-up phase, we compute the average AUC of classifiers trained to predict whether a sample will exhibit high memorization frequency. As shown in Table 10, the AUC values range from 73.78% on DEFAULT to 80.14% on SHOPPERS, indicating a strong alignment between early-phase memorization behavior and the final memorization frequency observed during training.

**Obs: Warm-up Memorization as a Reliable Filtering Signal.** These results indicate that early-stage (warm-up) memorization dynamics provide a robust signal for detecting memorization-prone samples. The consistently high AUC scores across datasets confirm that the warm-up labeling strategy effectively captures patterns linked to memorization, supporting its use as a reliable filtering criterion in DynamicCut.

### B.7 Experiments on Data Generation Quality

#### B.7.1 Feature Correlation Matrix Comparison

To assess whether DCM affects the structural fidelity of synthetic data, we compare the absolute divergence between the pairwise feature correlation matrices of the real and synthetic data. As shown in Figure 10, lighter colors indicate smaller divergences, implying better preservation of inter-feature relationships.

**Obs: Preservation of Feature Correlations after DCM Filtering.** Across all three datasets—ADULT, DEFAULT, and SHOPPERS—the heatmaps reveal that DCM-filtered models retain correlation structures that are visually comparable to, and in some cases slightly stronger than, those of the base TabDDPM. The minimal differences suggest that DCM's memorization filtering does not distort the underlying statistical dependencies among features.

These results demonstrate that DynamicCut can effectively reduce memorization without compromising the core structural quality of the generated data.

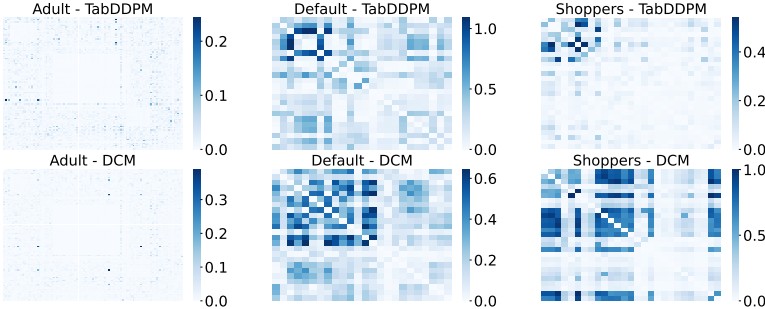

Figure 10: Heatmaps of the pair-wise column correlation of synthetic data v.s. the real data. The value represents the absolute divergence between the real and estimated correlations (the lighter, the better).

#### B.7.2 Shape Score

To further evaluate the quality of the synthetic data, we compare the shape score of each feature across datasets, as shown in Figure 11. The shape score measures the similarity between the marginal distributions of real and synthetic data, with higher values indicating better alignment.

**Obs: Preservation of Marginal Distributions after DCM Filtering.** Across all datasets—ADULT, DEFAULT, and SHOPPERS—the shape scores of TabDDPM and TabDDPM+DCM remain closely aligned for most features. In particular, for ADULT and DEFAULT, the scores exceed 0.95 for the majority of features, indicating that marginal distributions are well preserved after applying DCM. Although SHOPPERS exhibits higher feature variability, DCM still maintains competitive shape scores relative to the base model.

These results confirm that DynamicCut achieves effective memorization reduction without degrading the fidelity of per-feature distributions, thus preserving a key aspect of data generation quality.

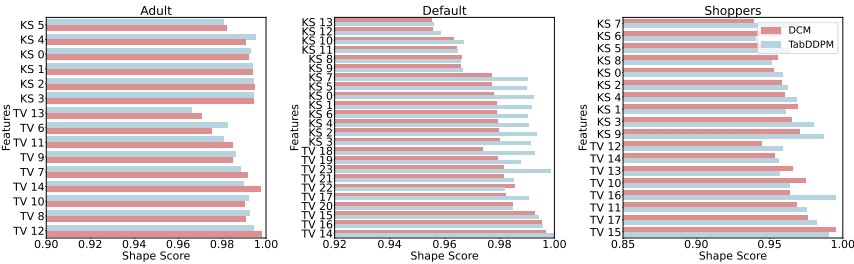

Figure 11: Shape score comparison for each feature in synthetic data generated by TabDDPM and TabD-DPM+DCM across multiple datasets.

## B.8 Computational Overhead of DynamicCut

The warm-up memorization monitoring stage does not introduce any additional forward or backward passes. The overhead mainly comes from tracking and aggregating per-sample Mem-AUC statistics during the first $T$ epochs. Specifically, collecting Mem-AUC values incurs a cost of $O(TN)$, and computing the top-$k$ statistics for each sample costs $O(TN \log N)$ in a naive implementation, or $O(TN)$ using a streaming top-$k$ scheme. In contrast, the overall training cost of the backbone model is $O(EN(C_{\text{fwd}}+C_{\text{bwd}}))$, where $E$ is the total number of training epochs and $C_{\text{fwd}}$ and $C_{\text{bwd}}$ denote the costs of one forward and backward pass, respectively. Since $T \ll E$, the additional overhead introduced by DynamicCut is small compared to the full training cost.

In practice, Mem-AUC does not need to be computed at every epoch. In our implementation, we compute it once every 10 epochs during the warm-up stage. Computing Mem-AUC once takes about 4 seconds on the *Default* dataset (30,000 rows), about 1 second on the *Shoppers* dataset (12,330 rows), and about 8 seconds on the *Adult* dataset (48,842 rows). This trend is consistent with the analysis above: the monitoring cost scales roughly linearly with the dataset size $N$, so larger datasets incur higher per-evaluation cost. With this setting, the total warm-up monitoring cost is about 240 seconds for *Default*, about 200 seconds for *Shoppers*, and about 400 seconds for *Adult*. Compared to the baseline training time (65 minutes for *Default*, 60 minutes for *Shoppers*, and 63 minutes for *Adult*), this corresponds to roughly 5.8%, 5.3%, and 9.6% extra time, respectively. Since this cost only appears in the early warm-up stage and does not add any forward or backward passes, the overall overhead remains modest in all cases.

## B.9 Average Trade-off Between Memorization Reduction and Generation Performance

As shown in Table 11, **DCM** consistently provides the strongest memorization reduction among all compared methods, both on the *Default* dataset and when averaging over all datasets and backbones. In particular, DCM achieves the highest average reduction on *Default* (8.78%) and a substantially larger reduction in the overall setting (13.47%), clearly outperforming SMOTE, TCM, and TCMP in terms of memorization mitigation.

Importantly, this stronger suppression of memorization does not come with a systematic degradation in generation quality. Across both *Default* and *All*, DCM remains competitive with the other methods on all evaluated quality metrics, and its performance is generally comparable to the best baseline in each column. This indicates that the gain in memorization reduction is not obtained by sacrificing overall sample quality.

More concretely, on *Default*, DCM attains the best MLE score and the highest Trend score, while staying close to TCM and TCMP on $\alpha$-Precision, $\beta$-Recall, Shape, C2ST, and DCR. When considering the overall average across all datasets and backbones, DCM again achieves the best or tied-best performance on several metrics, including MLE, $\alpha$-Precision, Trend, and C2ST, while remaining comparable on the remaining metrics. These results demonstrate that DCM offers a favorable trade-off: it more effectively reduces memorization, yet preserves the overall generation quality, and in several aspects even improves it.

Table 11: Average trade-off summary. We report (i) averaged memorization reduction (Improv. in Table 3), and (ii) averaged generation performance metrics. Averages are computed over three backbones (TabDDPM, CTGAN, TVAE) for Default, and over all datasets/backbones (Default, Shoppers, Adult × TabDDPM, CTGAN, TVAE; 9 settings) for All.

| Setting | Method | Mem. Ratio (Improv.%↑) | MLE (%↑) | α-Precision (%↑) | β-Recall (%↑) | Shape (%↑) | Trend (%↑) | C2ST (%↑) | DCR |
|---|---|---|---|---|---|---|---|---|---|
| Default | SMOTE | 4.38 | 74.52 | 79.61 | 26.14 | 89.33 | 46.58 | 60.79 | 49.73 |
| | TCM | 5.96 | 74.91 | 84.28 | 25.75 | 90.90 | 83.37 | 68.48 | 50.04 |
| | TCMP | 8.13 | 73.24 | 83.72 | 26.15 | 90.66 | 82.52 | 68.30 | 49.99 |
| | DCM | 8.78 | 75.11 | 83.06 | 25.40 | 89.92 | 84.30 | 66.81 | 49.24 |
| All | SMOTE | 5.94 | 84.13 | 80.07 | 29.48 | 84.71 | 70.64 | 57.85 | 50.36 |
| | TCM | 8.30 | 83.98 | 81.99 | 30.12 | 87.97 | 85.46 | 64.23 | 49.77 |
| | TCMP | 7.55 | 83.38 | 81.54 | 29.29 | 86.62 | 84.11 | 64.06 | 49.89 |
| | DCM | 13.47 | 84.22 | 82.13 | 29.01 | 86.59 | 85.05 | 64.86 | 49.53 |

## B.10 Additional Baselines

To better situate our method within existing memorization-mitigation strategies for tabular generative models, we extend our evaluation under the TABDDPM backbone by adding an additional data-augmentation baseline, **Mixup**, alongside the previously included baselines **SMOTE**, **TCM**, and **TCMP**. The results are summarized in Table 12.

**Obs.1: DCM achieves stronger memorization reduction than augmentation-based baselines.** Mixup provides a consistent but limited reduction in memorization across datasets, with improvements of 4.50% on DEFAULT, 12.50% on SHOPPERS, and 3.14% on ADULT. In contrast, our **DCM** achieves the strongest memorization mitigation on DEFAULT and SHOPPERS, reducing memorization by 18.31% and 22.73%, respectively, while remaining competitive on ADULT (15.29%), where **TCMP** achieves a slightly larger reduction (15.83%). This overall pattern reinforces our main finding that filtering a small set of samples selected by early memorization dynamics is a strong and reliable intervention, especially on datasets where memorization is more heterogeneous.

Table 12: Performance comparison of additional memorization-mitigation baselines. "DCM" denotes our proposed **DynamicCutMix**.

| Dataset | Methods | Mem. Ratio (%) ↓ | Improv. | MLE (%)↑ | α-Precision(%)↑ | β-Recall(%)↑ | Shape Score(%)↑ | Trend Score(%)↑ | C2ST(%)↑ | DCR(%) |
|---|---|---|---|---|---|---|---|---|---|---|
| Default | TabDDPM | $19.33 \pm 0.45$ | - | $76.79 \pm 0.69$ | $98.15 \pm 1.45$ | $44.41 \pm 0.70$ | $97.58 \pm 0.95$ | $94.46 \pm 0.68$ | $91.85 \pm 6.04$ | $49.12 \pm 0.94$ |
| | TabDDPM+Mixup | $18.46 \pm 0.71$ | 4.50% ↓ | $77.18 \pm 0.35$ | $93.20 \pm 4.16$ | $42.59 \pm 1.13$ | $95.34 \pm 1.79$ | $90.32 \pm 3.31$ | $92.59 \pm 2.82$ | $52.36 \pm 1.57$ |
| | TabDDPM+SMOTE | $17.46 \pm 0.51$ | 9.66% ↓ | $76.92 \pm 0.35$ | $91.19 \pm 0.68$ | $40.52 \pm 0.65$ | $94.89 \pm 1.46$ | $28.63 \pm 2.28$ | $72.73 \pm 0.69$ | $50.95 \pm 0.38$ |
| | TabDDPM+TCM | $16.76 \pm 0.47$ | 13.26% ↓ | $76.47 \pm 0.60$ | $97.30 \pm 0.46$ | $38.72 \pm 2.78$ | $97.27 \pm 1.74$ | $93.27 \pm 2.52$ | $94.72 \pm 3.87$ | $50.23 \pm 0.53$ |
| | TabDDPM+TCMP | $18.00 \pm 0.24$ | 6.88% ↓ | $76.92 \pm 0.17$ | $98.26 \pm 0.25$ | $41.92 \pm 0.52$ | $97.37 \pm 0.09$ | $91.42 \pm 1.15$ | $95.64 \pm 0.49$ | $49.75 \pm 0.32$ |
| | TabDDPM+DCM | $15.79 \pm 2.06$ | 18.31% ↓ | $76.68 \pm 0.76$ | $91.97 \pm 1.78$ | $37.89 \pm 0.84$ | $93.62 \pm 2.78$ | $89.30 \pm 3.02$ | $87.63 \pm 2.15$ | $48.17 \pm 2.59$ |
| Shoppers | TabDDPM | $31.37 \pm 0.31$ | - | $92.17 \pm 0.32$ | $93.16 \pm 1.58$ | $52.57 \pm 1.30$ | $97.08 \pm 0.46$ | $92.92 \pm 3.27$ | $86.74 \pm 0.63$ | $51.36 \pm 0.63$ |
| | TabDDPM+Mixup | $27.45 \pm 1.88$ | 12.50% ↓ | $91.44 \pm 1.37$ | $94.80 \pm 0.68$ | $51.72 \pm 1.05$ | $92.14 \pm 4.16$ | $89.31 \pm 3.91$ | $82.34 \pm 3.24$ | $46.85 \pm 5.81$ |
| | TabDDPM+SMOTE | $26.64 \pm 1.46$ | 15.07% ↓ | $89.96 \pm 0.95$ | $94.41 \pm 4.67$ | $45.22 \pm 3.26$ | $90.78 \pm 0.49$ | $83.09 \pm 2.47$ | $64.05 \pm 1.44$ | $51.94 \pm 1.52$ |
| | TabDDPM+TCM | $25.56 \pm 1.17$ | 18.51% ↓ | $92.17 \pm 0.26$ | $94.41 \pm 1.49$ | $50.05 \pm 1.59$ | $97.18 \pm 0.34$ | $93.95 \pm 0.51$ | $86.96 \pm 0.50$ | $47.52 \pm 1.81$ |
| | TabDDPM+TCMP | $28.51 \pm 0.35$ | 9.12% ↓ | $92.09 \pm 0.99$ | $93.43 \pm 1.65$ | $52.30 \pm 0.73$ | $97.31 \pm 0.22$ | $94.79 \pm 0.30$ | $87.02 \pm 2.04$ | $50.83 \pm 0.59$ |
| | TabDDPM+DCM | $24.24 \pm 2.83$ | 22.73% ↓ | $91.46 \pm 0.74$ | $93.25 \pm 3.87$ | $46.15 \pm 4.42$ | $92.32 \pm 5.00$ | $89.11 \pm 4.95$ | $80.56 \pm 4.81$ | $49.74 \pm 3.34$ |
| Adult | TabDDPM | $31.01 \pm 0.18$ | - | $91.09 \pm 0.07$ | $93.58 \pm 1.99$ | $51.52 \pm 2.29$ | $98.84 \pm 0.03$ | $97.78 \pm 0.07$ | $94.63 \pm 1.19$ | $51.56 \pm 0.34$ |
| | TabDDPM+Mixup | $30.04 \pm 0.41$ | 3.14% ↓ | $90.82 \pm 0.12$ | $95.78 \pm 0.68$ | $47.65 \pm 1.35$ | $98.02 \pm 1.08$ | $96.78 \pm 1.33$ | $93.65 \pm 3.59$ | $50.86 \pm 0.86$ |
| | TabDDPM+SMOTE | $28.98 \pm 0.78$ | 6.56% ↓ | $90.41 \pm 0.36$ | $94.93 \pm 1.72$ | $46.10 \pm 0.65$ | $93.40 \pm 1.12$ | $90.76 \pm 1.76$ | $80.75 \pm 0.84$ | $51.82 \pm 0.56$ |
| | TabDDPM+TCM | $27.55 \pm 0.19$ | 11.16% ↓ | $91.15 \pm 0.06$ | $94.97 \pm 0.06$ | $47.43 \pm 1.46$ | $98.65 \pm 0.03$ | $97.75 \pm 0.07$ | $85.61 \pm 16.03$ | $50.99 \pm 0.65$ |
| | TabDDPM+TCMP | $26.10 \pm 2.11$ | 15.83% ↓ | $90.54 \pm 0.17$ | $92.26 \pm 6.97$ | $43.49 \pm 3.74$ | $95.10 \pm 4.27$ | $91.50 \pm 6.53$ | $84.76 \pm 10.12$ | $50.68 \pm 0.89$ |
| | TabDDPM+DCM | $26.27 \pm 0.57$ | 15.29% ↓ | $90.81 \pm 0.16$ | $95.23 \pm 1.14$ | $45.76 \pm 1.02$ | $98.27 \pm 0.32$ | $96.71 \pm 0.86$ | $92.64 \pm 1.65$ | $50.79 \pm 0.46$ |

## C  Discussion

### C.1  Limitation Discussion

While our work provides a principled and effective approach to reducing memorization in tabular diffusion models, it has several limitations.

First, the effectiveness of DynamicCut depends on a warm-up phase that sufficiently exposes early memorization patterns. For models or datasets where such patterns emerge later, the current strategy may underperform or miss certain high-risk samples.

Second, the method introduces additional hyperparameters—such as the warm-up length, top-$k$ selection ratio, and filtering percentage—which may affect performance. Although we use consistent settings across datasets, the optimal configuration may vary depending on data scale, feature types, and training dynamics.

We leave these directions for future exploration.

### C.2  Future Work

Our study opens several promising directions for future research.

First, a deeper investigation into why certain samples are more prone to memorization is needed. In particular, exploring memorization at the feature level—such as which attributes contribute most to high memorization intensity—could improve interpretability and inform more targeted interventions.

Second, our current method uses a fixed warm-up phase to identify memorization-prone samples. Future work could explore adaptive or continuous memorization monitoring strategies that dynamically adjust to training progress or model uncertainty.

Finally, while it is widely observed in practice that some samples are memorized much more frequently than others, a clear theoretical explanation is still missing. This behavior is closely related to data heterogeneity, and is connected to prior works Ghorbani & Zou (2019); Shrivastava et al. (2016); Kumar et al. (2010) on hard samples and data valuation. In future work, we plan to further study this problem and develop a theoretical framework to explain why certain samples are more likely to be memorized, which could help connect empirical findings with theory and guide more principled mitigation strategies.

## D  Experimental Details

We implement DynamicCut and DynamicCutMix along with all baseline methods using the PyTorch framework. All models are trained using the Adam optimizer. Our code is available at `https://github.com/fangzy96/DynamicCut`

### D.1  Baselines of Data Augmentation

This section outlines the data augmentation methods evaluated in this work, including SMOTE, TabCutMix (TCM), and TabCutMixPlus (TCMP).

- **SMOTE**: The Synthetic Minority Oversampling Technique (SMOTE) Chawla et al. (2002) is a foundational oversampling method initially proposed to address class imbalance in datasets with continuous variables. It generates new minority class samples by interpolating feature values between existing samples and their $k$ nearest neighbors. Although SMOTE is effective in enhancing diversity and reducing overfitting in fully numerical datasets, its design does not account for categorical attributes. Direct application to mixed-type data can therefore produce invalid or nonsensical feature combinations that violate feature semantics.

- **Mixup**: Mixup Zhang (2017) is a data augmentation method that generates new training examples by interpolating between pairs of samples and their labels. Although it is simple and has been shown to be effective in many settings, it implicitly enforces a linear interpolation in the feature space.

- **TabCutMix (TCM)**: TabCutMix (TCM) Fang et al. (2024) adapts the CutMix concept from computer vision to the tabular domain. TCM generates augmented samples by selectively replacing subsets of features between two instances from the same class, guided by a binary mask sampled from a Bernoulli distribution. The mask controls the replacement ratio $\lambda$, enabling partial feature replacement. By promoting intra-class variability while preserving label integrity, TCM improves generalization and reduces overfitting in mixed-type tabular learning tasks.

- **TabCutMixPlus (TCMP)**: TabCutMixPlus (TCMP) Fang et al. (2024) builds upon TCM by introducing dependency-aware feature augmentation. TCMP first performs feature clustering based on domain-relevant correlation metrics to identify groups of highly related features. Augmentation is then performed at the cluster level, ensuring that structurally or semantically related features are swapped together. This strategy helps maintain intra-cluster consistency, mitigates the risk of generating unrealistic data points, and enhances both diversity and robustness in the augmented dataset.

## D.2 Baselines of Tabular Data Generation

This section describes the baseline generative models used in our study. These methods represent widely recognized approaches for synthesizing structured tabular data with mixed numerical and categorical attributes.

- **CTGAN** Xu et al. (2019): CTGAN is a generative adversarial network (GAN) designed to address the unique challenges of tabular data generation. To better capture complex, skewed, or multimodal numerical feature distributions, CTGAN employs mode-specific normalization based on quantile transformation. Additionally, it leverages a conditional sampling strategy that focuses on rare categories when generating new samples. This conditional mechanism helps balance class distributions and promotes diversity across different feature values in the generated data.

- **TVAE** Xu et al. (2019): TVAE extends the variational autoencoder (VAE) framework to handle tabular data. Like CTGAN, it applies mode-specific normalization and conditional generation techniques, but models data through a latent probabilistic structure by optimizing the Evidence Lower Bound (ELBO). This allows TVAE to learn a meaningful latent representation that captures dependencies between numerical and categorical features, facilitating the generation of realistic and diverse synthetic samples while maintaining scalability.

- **TabDDPM** Kotelnikov et al. (2023): TabDDPM adapts denoising diffusion probabilistic models (DDPMs) for tabular data with mixed feature types. It applies continuous-time stochastic differential equations (SDEs) to latent representations of tabular samples, enabling score-based generative modeling. To jointly model numerical and categorical features, TabDDPM incorporates specialized feature embeddings and masking strategies within a unified diffusion framework. This design supports both unconditional and conditional sample generation, offering improved training stability and sample quality.

## D.3 Quantifying Memorization

Generative models risk overfitting when they overly replicate samples from their training data instead of producing diverse, novel instances representative of the full underlying data distribution. While retaining some level of fidelity is beneficial in certain use cases—such as high-precision applications—over-memorization can hinder diversity, introduce privacy risks, and compromise the model's ability to generalize. This issue is particularly relevant in tabular data scenarios, where direct reproduction of training records may lead to sensitive information leakage or limit the usefulness of generated data.

**Relative Distance Ratio Criterion.** To systematically evaluate the extent of memorization, we introduce the *relative distance ratio* as a sample-level diagnostic. Given a generated sample $x$ and the training

dataset $\mathcal{D}$, the distance ratio $r(x)$ is computed as:

$$r(x) = \frac{d(x, \text{NN}_1(x, \mathcal{D}))}{d(x, \text{NN}_2(x, \mathcal{D}))},$$

where $d(\cdot, \cdot)$ is a distance function in the data space, $\text{NN}_1(x, \mathcal{D})$ is the closest training sample to $x$, and $\text{NN}_2(x, \mathcal{D})$ is the second closest. A lower $r(x)$ indicates that $x$ is unusually close to a single training point compared to others, signaling potential memorization.

Following best practices from the domain of image synthesis, we label a generated sample as *memorized* if its distance ratio falls below a threshold of $\frac{1}{3}$.

**Memorization AUC.** To capture how memorization varies over a range of sensitivity thresholds, we introduce the *Memorization Area Under the Curve (Mem-AUC)*. This metric aggregates the memorization ratio across all threshold values $\tau \in [0, 1]$, providing a comprehensive view of memorization intensity:

$$\text{Mem-AUC} = \int_0^1 \text{Mem. Ratio}(\tau) \, d\tau,$$

where Mem. Ratio$(\tau)$ denotes the fraction of generated samples with $r(x) < \tau$ for a given threshold $\tau$.

Higher Mem-AUC values indicate stronger overall memorization tendencies, while lower values reflect better generalization and less replication of the training data. Together, the memorization ratio and Mem-AUC offer both fixed-threshold and threshold-sweeping perspectives on the memorization behavior of generative models.

**Memorization Ratio.** We define the *memorization ratio* as the fraction of generated samples that meet this memorization condition:

$$\text{Mem. Ratio} = \frac{1}{|\mathcal{G}|} \sum_{x \in \mathcal{G}} \mathbb{I}\left(r(x) < \frac{1}{3}\right),$$

where $\mathcal{G}$ represents the set of generated instances and $\mathbb{I}(\cdot)$ is an indicator function. This score provides a single-point estimate of the prevalence of memorization in the generated dataset.

**Tracking Memorization Dynamics.** To move beyond a single, static memorization snapshot, we follow each real training sample *through epoch* and document when and how often it becomes memorized, forgotten, and re-memorized. Below we introduce the exact notation and event definitions that make this temporal analysis reproducible. We denote $\mathcal{D} = \{x_r\}_{r=1}^{|\mathcal{D}|}$ as training set, $\tau$ as a fixed memorization threshold (we adopt $\tau = \frac{1}{3}$), and $\mathcal{G}_e = \{x_{g,e}^{(j)}\}_{j=1}^{|\mathcal{G}_e|}$ as the samples generated *after* epoch $e \in \{1, \ldots, E\}$.

For every real sample $x_r$ and epoch $e$, we record a binary flag

$$\mathbb{1}_e(x_r) = \mathbb{1}\Big[\exists \, x \in \mathcal{G}_e : r(x) < \tau \ \wedge \ \text{NN}_1(x, \mathcal{D}) = x_r\Big],$$

where $\text{NN}_1(\cdot, \mathcal{D})$ returns the nearest neighbour in the training set. Intuitively, $\mathbb{1}_e(x_r) = 1$ means that, at epoch $e$, at least one generated sample sits "too close" to $x_r$, signalling potential leakage of that sample.

**Event Definitions.** We translate the per-epoch indicators into discrete events that mirror our informal reasoning about memorization behavior.

$$\text{FirstMem}(x_r) = \min\{\, e : \mathbb{1}_e(x_r) = 1\}$$

$$\text{(first epoch } x_r \text{ is memorized)},$$

$$\text{Forget}(x_r) = \big\{\, e : \mathbb{1}_{e-1}(x_r) = 1 \wedge \mathbb{1}_e(x_r) = 0\big\}$$

$$\text{(moments when } x_r \text{ stops being memorized)}.$$

If $\text{FirstMem}(x_r)$ is undefined, the model never memorizes $x_r$.

Table 13: Statistics of datasets. *Num* indicates the number of numerical columns, and *Cat* indicates the number of categorical columns.

| Dataset | #Rows | #Num | #Cat | #Train | #Validation | #Test | Task |
|---------|-------|------|------|--------|-------------|-------|------|
| Adult | 48,842 | 6 | 9 | 28,943 | 3,618 | 16,281 | Classification |
| Default | 30,000 | 14 | 11 | 24,000 | 3,000 | 3,000 | Classification |
| Shoppers | 12,330 | 10 | 8 | 9,864 | 1,233 | 1,233 | Classification |
| Cardio | 70,000 | 5 | 7 | 44,800 | 11,200 | 14,000 | Classification |
| Wilt | 4,889 | 5 | 1 | 3,096 | 775 | 968 | Classification |

**Count Statistics.** A single event seldom tells the full story. We therefore accumulate events into counts that capture temporal behaviors and volatility.

$$\text{CumMemCnt}(x_r) = \sum_{e=1}^{E} \mathbb{1}_e(x_r)$$

total epochs during which $x_r$ is memorized,

$$\text{CumForgetCnt}(x_r) = |\text{Forget}(x_r)|$$

number of "un-memorization" transitions.

High *CumMemCnt* indicates long-lasting leakage; a high *CumForgetCnt* suggests unstable or oscillatory behavior.

**Instance-level Mem-AUC.** To capture *how strongly* a sample $x_r$ is memorized—independent of an arbitrary threshold—we integrate the conditional memorization probability over every possible threshold:

$$\text{Mem-AUC}(x_r) \; = \; \int_0^1 \Pr\big[r(x) < \tau \,\big|\, \text{NN}_1(x, \mathcal{D}) = x_r\big] \, d\tau.$$

Here, the probability is estimated over all $x \in \mathcal{G}_e$ and all epochs $e$. Larger values imply stronger, more persistent memorization. Such instance-level Mem-AUC provides a quantitative way to detect high-memorization samples based on temporal and memorization strength signal, such as top-10 % (*high-risk*) and bottom-10 % (*low-risk*) Mem-AUC subsets.

In a nutshell, these precise definitions underpin our analysis and pinpoint when memorization first emerges, how long it persists, and how often the model successfully "forgets" individual records. The temporal lens complements the static memorization ratio and provides a new dimension for memorization mitigation throughout training.

## D.4   Datasets

We conduct our experiments on three widely recognized benchmark datasets sourced from the UCI Machine Learning Repository. These datasets span diverse application domains, including income prediction, credit risk assessment, and online customer behavior analysis. Table 13 provides an overview of their key characteristics.

**Detailed Dataset Summaries.**

- **Adult Dataset**[2]: Also known as the Census Income dataset, this dataset contains demographic and employment information from the 1994 U.S. Census. The binary classification task aims to predict whether an individual's annual income exceeds $50,000. It consists of 48,842 records with features such as age, education, occupation, marital status, and work class. This dataset is commonly used in fairness, socio-economic, and income classification studies.

---

[2]https://archive.ics.uci.edu/dataset/2/adult

- **Default Dataset**[3]: This dataset includes 30,000 instances of credit card clients in Taiwan, featuring financial and demographic information collected between April and September 2005. The goal is to predict the likelihood of a client defaulting on their credit card payment in the next billing cycle. Key attributes include credit limit, payment history, bill amounts, and personal demographics, making it particularly relevant for financial risk modeling and credit scoring research.

- **Shoppers Dataset**[4]: Collected from user sessions on e-commerce platforms, this dataset consists of 12,330 records detailing user interaction patterns. Features include the number of pages visited, session durations, and various behavioral indicators. The binary target indicates whether a session ended with a purchase. This dataset serves as a benchmark for studying customer behavior, website optimization, and purchase intent prediction.

- **Wilt Dataset**[5]: This dataset originates from high-resolution remote sensing imagery and is designed for binary classification of vegetation health. The task involves distinguishing between healthy land cover (label 'n') and areas exhibiting signs of disease or stress (label 'w'), based on $4,889$ observations. Each instance is described by spectral and textural attributes extracted from Quickbird satellite data, including metrics like GLCM mean texture, normalized color bands (green, red, NIR), and statistical features such as the standard deviation of the panchromatic band. The dataset is notably imbalanced, with only 74 samples labeled as diseased, presenting additional challenges for generative modeling and minority class preservation.

- **Cardio Dataset**[6]: This dataset contains anonymized health profiles of $70,000$ individuals, collected for the purpose of predicting cardiovascular disease risk. The feature set spans 11 variables covering demographic information (e.g., age, gender, height, weight), clinical measurements (e.g., systolic and diastolic blood pressure, cholesterol levels, glucose concentration), and lifestyle indicators (e.g., smoking, alcohol use, physical activity). The binary target indicates whether a subject has been diagnosed with cardiovascular disease. The dataset is moderately imbalanced and has been widely used in evaluating tabular generative and classification models.

In Table 13, The column "# Rows" represents the number of records in each dataset, while "# Num" and "# Cat" indicate the number of numerical and categorical features (including the target feature), respectively. Each dataset is divided into training, validation, and testing sets for machine learning efficiency experiments. For the Adult dataset, which includes an official test set, we use this set directly for testing, and split the training set into training and validation sets with a ratio of 8 : 1. For the remaining datasets, the data is partitioned into training, validation, and test sets with a ratio of 8 : 1 : 1, ensuring consistent splitting using a fixed random seed.

---

[3]https://archive.ics.uci.edu/dataset/350/default+of+credit+card+clients
[4]https://archive.ics.uci.edu/dataset/468/online+shoppers+purchasing+intention+dataset
[5]https://archive.ics.uci.edu/dataset/285/wilt
[6]https://www.kaggle.com/datasets/sulianova/cardiovascular-disease-dataset

