# OpenReview forum: "A Closer Look on Memorization in Tabular Diffusion Model: A Data-Centric Perspective"
_TMLR — Accepted by TMLR_

### Review · Reviewer_jGGV · 2025-12-12

**Summary Of Contributions:**

In the manuscript, the authors present a data-centric and sample-level analysis of memorization dynamics in tabular diffusion models, revealing that memorization is highly uneven and dominated by a small subset of training samples. By introducing fine-grained metrics such as per-sample memorization counts, first-memorization time, and memorization intensity, the authors show that memorization-prone samples are typically memorized early and exhibit strong early-stage signals. Leveraging these insights, the paper proposes DynamicCut, a simple, model-agnostic mitigation strategy that identifies and removes high-risk samples based on early training dynamics, and DynamicCutMix, which combines targeted filtering with feature-level data augmentation. The authors carry out extensive experiments across multiple datasets and generative models to demonstrate that the proposed methods substantially reduce memorization while preserving data quality.

**Audience:**

Yes

**Audience Explanation:**

Yes, at least some individuals in TMLR’s audience would be interested in the findings of this paper. The work provides novel insights into memorization in generative tabular models, a topic of broad relevance to researchers in machine learning, data privacy, and generative modeling.

**Broader Impact Concerns:**

No concerns.

**Claims And Evidence:**

Yes

**Claims Explanation:**

The claims in the paper are supported by clear, convincing, and well-structured evidence. The authors provide a thorough empirical analysis across multiple benchmark tabular datasets and generative models, demonstrating that memorization is concentrated in a small subset of training samples and arises early during training. These claims are substantiated through well-defined memorization metrics, ablation studies (e.g., targeted vs. random sample removal), temporal analyses of memorization dynamics, and extensive quantitative evaluations showing that the proposed DynamicCut and DynamicCutMix methods reduce memorization with minimal impact on data quality and downstream performance.

**Requested Changes:**

While the paper is well written and contains many strengths, the following changes could make its findings more robust and broadly impactful.

1. Clarify Practical Implications (Important): While the empirical findings are strong, I would appreciate a clearer discussion of how practitioners might use DynamicCut in real deployments of tabular generative models (e.g., healthcare or finance). In particular, guidance on when sample removal is preferable to augmentation-based approaches would strengthen the paper’s practical relevance.
2. Sensitivity to Hyperparameters (Important): A short discussion or additional experiment examining sensitivity to the pruning ratio p and early-epoch window choices would make the method easier to reproduce and tune in practice.
3. Broader Comparison with Recent Baselines (Minor): The experimental evaluation is thorough, but including comparisons with a broader set of recent memorization-mitigation techniques (especially those developed for diffusion models in other domains) would help better situate the proposed method within the growing literature.

---

> ### Author Response · Authors · 2026-02-17
> **Response to Reviewer jGGV**
>
> We thank the reviewer for the careful and constructive feedback. The comments are very helpful and have helped us further solidify the paper. We have taken each point seriously and addressed all concerns in detail below.
>
> 1. Clarify Practical Implications (Important).
>
> DynamicCut is particularly suitable for scenarios where memorization is highly heterogeneous across samples, since in this regime removing a small fraction of carefully selected samples can already resolve a large portion of the memorization problem. This is exactly the setting we observe in practice. For example, Table 1 provides a concrete illustration on the Adult dataset, which is a representative finance dataset. When removing only 5% of the training samples selected by our method (“Label 5%”), the memorization ratio is reduced by 16.63% relative to the baseline. In contrast, removing the same number of samples at random (“Random 5%”) barely changes the memorization ratio, with only a −1.56% relative change. This large gap shows that memorization is not uniformly distributed and that a small, high-risk subset of samples contributes disproportionately to the problem.
>
> From a practical perspective, this explains when sample removal is preferable to augmentation-based approaches. In heterogeneous datasets, targeting and removing a small set of high-memorization samples is a more direct and efficient intervention than globally augmenting or reweighting the data, since most of the memorization risk is concentrated in a small subset. Our experimental evaluation spans multiple realistic domains, including finance (Adult, Default), healthcare (Cardio), e-commerce behavior (Shoppers), and remote sensing with severe imbalance (Wilt), which suggests that this scenario is common in real-world tabular data.
>
> 2. Sensitivity to Hyperparameters (Important).
>
> In our method, the two key hyperparameters that control the behavior of DynamicCut are (i) the pruning ratio p used to select high-memorization samples and (ii) the length of the warm-up window that defines early memorization dynamics. We already include dedicated ablation studies for both of these choices in the paper and further extend this analysis to strengthen the robustness discussion.
>
> First, for the pruning ratio p, Appendix B.1 and Table 6 compare a wide range of settings (1%, 3%, 5%, 10%, and 20%). The results show a clear and stable trend: very small values are too conservative and lead to limited memorization reduction, while overly large values become less precise and yield weaker or less stable improvements. A moderate choice (p = 10\%) consistently achieves the best balance between memorization reduction and utility across datasets. This demonstrates that the effect of DynamicCut is not tied to a single fragile choice of p.
>
> Second, regarding the warm-up window, we treat it as a data-dependent design choice rather than a fixed universal hyperparameter, since different datasets can exhibit different training dynamics. To make this explicit, we have added a dedicated ablation study in Appendix B.2 and Table 7, where we vary the warm-up window length (e.g., 600/700/800 for Default and 2000/2100/2200 for Shoppers) and evaluate the resulting changes in both memorization mitigation and generation quality. The results show that the optimal window length is indeed dataset-dependent, and that using an overly long window tends to weaken memorization reduction. At the same time, the performance remains stable under reasonable variations of the window length, indicating that DynamicCut is not sensitive to a particular tuned choice and is robust in practice.

---

> > ### Author Response · Authors · 2026-02-17
> > **Response to Reviewer jGGV**
> >
> > 3. Broader Comparison with Recent Baselines (Minor).
> >
> > In this revision, we have taken a concrete step in this direction by adding an additional data-augmentation baseline, Mixup, together with the previously included SMOTE, TCM, and TCMP. The new results are reported in Appendix B.10 and Table 12. This extends the comparison to a stronger and widely used augmentation strategy and provides a more comprehensive view of how DynamicCutMix compares to representative data-level and training-time interventions.
> >
> > At the same time, we would like to clarify that many recent memorization-mitigation methods developed for diffusion models in other domains are often built on domain-specific assumptions, architectures, or evaluation protocols, which makes a direct and fair transfer to tabular diffusion models non-trivial. For this reason, we focus on baselines that are broadly applicable to tabular data and can be evaluated under the same experimental protocol. In particular, data augmentation and training-time mixing or filtering strategies remain the most practical and comparable choices in the current tabular generative modeling literature. By adding Mixup, which is a widely used and domain-agnostic augmentation method, and systematically comparing against SMOTE, TCM, and TCMP, we aim to provide a fair and representative comparison set in this setting. The results in Table 12 show that DCM remains among the strongest methods for memorization mitigation under these comparable and practically relevant baselines.

---

> ### Comment · Action_Editor_hw9o · 2026-03-03
> **Reviewer Reminder: Submit Your Official Recommendation**
>
> Hello Reviewer
>
> The authors have submitted their responses and updated the paper.
> This is a reminder to submit your official recommendation for this paper!

---

### Review · Reviewer_WSrJ · 2026-01-22

**Summary Of Contributions:**

The paper presents a data-centric investigation into memorization dynamics in tabular diffusion models, addressing an underexplored yet important problem at the intersection of synthetic data generation and privacy.

**Key Contributions:**

1. **Novel empirical analysis**: The paper provides the first fine-grained, sample-level analysis of memorization in tabular diffusion models, revealing that memorization follows a heavy-tailed distribution where a small subset of training samples accounts for disproportionate memorization.

2. **Temporal dynamics characterization**: The authors track memorization events across training epochs, finding that high-memorization samples tend to be memorized earlier and exhibit stronger memorization intensity (Mem-AUC) in early training stages.

3. **DynamicCut method**: A lightweight, model-agnostic filtering strategy that identifies and removes memorization-prone samples based on early-stage Mem-AUC dynamics, requiring only partial training trajectories.

4. **DynamicCutMix**: Integration of DynamicCut with existing data augmentation (TabCutMix) for enhanced memorization mitigation.

5. **Cross-model transferability**: Demonstration that memorization-prone samples identified from one model (e.g., TabDDPM) can be effectively used to reduce memorization in other architectures (CTGAN, TVAE).

**Key Strengths:**
- Well-motivated problem with clear privacy implications in sensitive domains
- Comprehensive empirical analysis across multiple datasets and generative models
- Simple, practical, and model-agnostic method
- Extensive ablation studies and hyperparameter analysis
- Strong experimental design with multiple evaluation metrics covering both memorization and utility

**Key Weaknesses:**
- Limited theoretical grounding for why certain samples are memorized more
- Computational overhead of the warm-up phase not thoroughly discussed

**Audience:**

Yes

**Audience Explanation:**

The paper addresses a timely and important problem that would interest multiple research communities.

**Claims And Evidence:**

Yes

**Claims Explanation:**

The paper provides substantial empirical evidence supporting its main claims.

**Requested Changes:**

1. **Computational cost analysis**: The paper should report the computational overhead introduced by the warm-up phase monitoring. How does the cost scale with dataset size and number of epochs?

2. **Clearer algorithm presentation**: Section 5.3 describes the algorithm but lacks a formal algorithm block.

---

> ### Author Response · Authors · 2026-02-17
> **Response to Reviewer WSrJ**
>
> We thank the reviewer for the careful and constructive feedback. The comments are very helpful and have helped us further solidify the paper. We have taken each point seriously and addressed all concerns in detail below.
>
> 1. Computational cost analysis.
>
> We have added a new subsection (B.8 Computational Overhead of DynamicCut) to report the computational overhead of the warm-up memorization monitoring stage. This stage does not introduce any additional forward or backward passes. The overhead mainly comes from tracking and aggregating per-sample Mem-AUC statistics during the first $T$ epochs. The cost scales linearly with the dataset size and the number of monitored epochs: collecting Mem-AUC values costs $O(TN)$, and computing the top-$k$ statistics costs $O(TN \log N)$ in a naive implementation or $O(TN)$ with a streaming top-$k$ scheme. In contrast, the overall training cost is $O(EN (C_{\text{fwd}} + C_{\text{bwd}}))$, and since $T \ll E$, the additional overhead is small compared to full training.
>
> In practice, Mem-AUC is not computed at every epoch. We compute it once every 10 epochs during the warm-up stage. Computing Mem-AUC once takes about 4 seconds on the Default dataset (30,000 rows), 1 second on the Shoppers dataset (12,330 rows), and 8 seconds on the Adult dataset (48,842 rows). This trend matches the analysis above: the monitoring cost scales approximately linearly with the dataset size $N$, so larger datasets incur higher per-evaluation cost. With this setting, the total warm-up monitoring cost is about 240 seconds for Default, 200 seconds for Shoppers, and 400 seconds for Adult. Compared to the baseline training time (65 minutes for Default, 60 minutes for Shoppers, and 63 minutes for Adult), this corresponds to roughly 5.8\%, 5.3\%, and 9.6\% extra time, respectively. Since this cost only appears in the early warm-up stage and does not add any forward or backward passes, the overall overhead remains modest in all cases.
>
> 2. Clearer algorithm presentation.
>
> We have now added a formal algorithm block in the current Section 5.2 (Algorithm 1) to present the method in a clearer and more precise manner. The new algorithm block summarizes the full procedure and aligns with the textual description in the section.
>
> 3. Limited theoretical grounding for why certain samples are memorized more.
>
> Providing a full theoretical explanation of why certain samples are memorized more than others is an open question. In this work, we deliberately do not aim to resolve this question. Instead, our goal is to take a data-centric perspective: to empirically measure memorization, to show that it is highly heterogeneous across samples, and to demonstrate that a simple filter-based intervention guided by training dynamics can effectively mitigate memorization in practice.
>
> To make this scope explicit, we have added a dedicated discussion in Section C.2 (Future Work) in the revised manuscript, where we relate this question to data heterogeneity and prior work on hard samples and data valuation, such as Data Shapley and curriculum or hard example mining, and add the corresponding citations. The key contribution of this paper is therefore not a theoretical model of memorization, but an empirical characterization and a practical mitigation strategy. Developing a principled theoretical account of why a small subset of samples is disproportionately memorized is a complementary and important direction that we leave for future work.

---

> > ### Comment · Action_Editor_hw9o · 2026-03-03
> > **Reviewer Reminder: Please submit your official recommendation**
> >
> > Hello Reviewer
> >
> > The authors have submitted their responses and updated the paper.
> > This is a reminder to submit your official recommendation for this paper!

---

> > > ### Comment · Reviewer_WSrJ · 2026-03-11
> > >
> > > Hello AE,
> > > Apologies for the delay in submitting the recommendation. I have submitted my official recommendation.
> > > Thanks

---

### Review · Reviewer_Eg4g · 2026-02-03

**Summary Of Contributions:**

This paper provides a detailed empirical investigation of memorization in tabular diffusion models from a data-centric perspective. The authors introduce the notion of high-memorization samples, study their appearances and temporal dynamics during training, and propose mitigation strategies based on conditional and dynamic control mechanisms. Through extensive experiments on multiple tabular datasets, the paper analyzes how memorization correlates with training dynamics, forgetting behavior, and downstream generation quality. The work aims to improve understanding of memorization beyond aggregate metrics by focusing on sample-level behavior over time.

**Audience:**

Yes

**Audience Explanation:**

The topic of memorization in generative models, especially diffusion models applied to tabular data, is timely and relevant to the TMLR audience. Researchers interested in data-centric analysis, generalization, privacy, and robustness of generative models would find the empirical insights and methodological focus valuable. The paper opens interesting directions for understanding memorization dynamics at the sample level, which is of broader relevance beyond tabular diffusion models.

**Broader Impact Concerns:**

No broader impact concerns.

**Claims And Evidence:**

Yes

**Claims Explanation:**

Yes but partially.

While the paper presents a substantial amount of experimental evidence and visualizations, several key claims rely on heuristic choices, speculative interpretations, or insufficiently justified experimental design decisions.

In particular, the definition and selection of high-memorization samples relies on thresholds (e.g., 5%, 10%, 20%) that are not well aligned with the empirical distribution of sample frequencies, where samples with frequency greater than 3 typically constitute less than 5% of the dataset. This mismatch raises concerns about whether the selected thresholds meaningfully capture memorization or instead introduce arbitrary groupings.

Moreover, in Figure 1, authors use overly wide scales. This makes it hard to see how different performances compare to each other.

Also, visual evidence (e.g., Figure5(A) comparisons in Section 5.2) does not clearly support the stated conclusions without further quantitative analysis.

**Requested Changes:**

1. Clarify and justify the definition of High-Memorization Samples.

The current definition relies on selecting the top 5%, 10%, or 20% of samples by frequency. However, empirical frequency distributions indicate that samples with frequency greater than 3 typically account for less than 5% of the data. This raises concerns about whether larger thresholds (10%, 20%) meaningfully represent memorization. The authors should provide a principled justification for these thresholds.

2. Reframe or validate Section 4.2, Observation 2 (Possible Causes of Early Memorization).

This subsection is labeled as an “observation,” but the discussion is largely speculative and causal claims are not directly supported by controlled experiments. The authors should either provide additional empirical validation or clearly reframe this part as a hypothesis or conjecture rather than an observation.

3. Reconsider the placement of Section 5.2 (Temporal Dynamics of Forgetting).

The content of Section 5.2 is analytical and empirical rather than methodological. Its current placement within the Methodology section is confusing. The section would be better positioned within the experimental analysis or results section to improve the logical flow of the paper.

4. Strengthen evidence for Section 5.2, Observation 1 (Early Forgetting of High-Memorization Samples).

The visual differences shown in the corresponding figures appear relatively small, making it difficult to assess the statistical or practical significance of the claim. Additional quantitative metrics, statistical tests, or clearer visualizations are needed to convincingly support this observation.

5. Provide justification and comparison for algorithmic hyperparameter choices.

Several hyperparameters in the proposed algorithms are fixed without sufficient explanation. A comparison with alternative settings or sensitivity analysis would help demonstrate that the reported effects are not artifacts of particular hyperparameter choices.

6. Revise Figure 1 to ensure fair comparison.

In Figure 1, the axes for metrics such as MLE and Trend appear to use overly wide scales, which visually compresses performance differences and makes comparisons misleading. The authors should use consistent and appropriately scaled axes to reflect the true magnitude of differences.

7. More thoroughly analyze the trade-off between memorization and performance.

The balance between memorization reduction and generation performance is not sufficiently explored. For example, Table 3 shows that on Default dataset, DCM performs worse than TCM and TCMP according to α-Precision, yet this result is not mentioned in analysis. A more thorough discussion is needed to clarify when reduced memorization comes at a tangible performance cost and how this trade-off should be interpreted.

---

> ### Author Response · Authors · 2026-02-17
> **Response to Reviewer Eg4g**
>
> We thank the reviewer for the careful and constructive feedback. The comments are very helpful and have helped us further solidify the paper. We have taken each point seriously and addressed all concerns in detail below.
>
> 1. Clarify and justify the definition of High-Memorization Samples.
>
> We emphasize that the goal of this paper is not to propose a theoretically optimal cutoff for defining memorization, but to show that a filter-based intervention guided by memorization dynamics is effective in practice and admits a robust empirical default across datasets. However, to directly address the concern, we have added a more comprehensive ablation study that evaluates a wide range of pruning ratios, from 1%, 3%, and 5% to 10% and 20%, to explicitly demonstrate the robustness and trade-offs of this choice (Appendix B.1.1). Importantly, our selection is based on a continuous memorization score derived from warm-up Mem-AUC dynamics and a ranking-based percentile criterion, rather than a hard threshold on discrete frequency counts.
>
> In this context, we clarify that DynamicCut does not define high-memorization samples using a hard frequency cutoff, and the frequency plot is not used as the selection rule. The observation that samples with frequency greater than 3 account for less than 5% of the data is expected and simply reflects a typical long-tailed memorization pattern, where only a small fraction of samples are replicated many times. This figure is included to illustrate the long-tail phenomenon and to help visualize the distribution, not to determine which samples are filtered. Instead, DynamicCut identifies high-memorization samples based on warm-up memorization dynamics measured by Mem-AUC. Each training sample is assigned a continuous memorization score aggregated from its early-epoch Mem-AUC values, and we remove a top-p fraction of samples according to this ranking. This ranking-based, percentile criterion is more stable and comparable across settings than using an absolute frequency threshold, since the scale of memorization frequency can vary substantially across datasets, models, and training configurations. In contrast, a fixed cutoff such as “frequency > k” would be brittle and not transferable across scenarios.
>
> However, to directly address the concern about whether larger thresholds (e.g., 10% or 20%) meaningfully represent memorization, we added a finer-grained ablation with p = 1%, 3%, 5%, 10%, and 20% (Appendix B.1.1). The results show that very small thresholds (1% and 3%) are too conservative and lead to limited memorization reduction, while overly large thresholds (20%) become less precise and yield weaker or less stable improvements. A moderate threshold around 10% consistently achieves the best trade-off between memorization mitigation and generation quality across datasets, which is why we use p = 10% as the default setting.
>
> 2. Reframe or validate Section 4.2, Observation 2 (Possible Causes of Early Memorization).
>
> We agree that the previous wording of Observation 2 could be interpreted as making causal claims, while our intention was to summarize an empirical pattern observed in the memorization trajectories. In the revised manuscript, we have refined Observation 2 to focus strictly on a directly observable result, namely the difference in convergence speed between the Top 5% and Non-Top groups shown by the cumulative memorization curves, without making causal statements. This revision improves the precision of the presentation while keeping the core empirical findings unchanged.
>
> 3. Reconsider the placement of Section 5.2 (Temporal Dynamics of Forgetting).
>
> We agree that Section 5.2 (Temporal Dynamics of Forgetting) is analytical and empirical in nature rather than methodological. In the revised manuscript, we have moved the “Temporal Dynamics of Forgetting” analysis out of the Methodology section and placed it in 4.3 within the preliminary analysis. We keep memorization and forgetting as two parallel subsections to clearly present these two complementary phenomena, while improving the overall logical flow of the paper.

---

> > ### Author Response · Authors · 2026-02-17
> > **Response to Reviewer Eg4g**
> >
> > 4. Strengthen evidence for Section 5.2, Observation 1 (Early Forgetting of High-Memorization Samples).
> >
> > Although the gap between the curves may look small by visual inspection, this is mainly a perceptual issue of the plot scale and does not imply that the difference is negligible. In fact, this separation is already sufficient to design an algorithm that distinguishes high-memorization samples from the rest based on their training dynamics. To move beyond visual impressions, we quantify this difference using the Kolmogorov–Smirnov (KS) test and also measure the corresponding separation on the time (epoch) axis.
> >
> > For the Top high-memorization samples, the maximum vertical separations are 8.36% (Shoppers, p ≈ 0), 4.66% (Default, p = 6.4e−99), and 1.28% (Adult, p = 2.5e−11), which indicates statistically significant differences between the two distributions in all three datasets. The extremely small p-values are expected here, since the comparison involves many samples and the two groups exhibit a consistent and systematic shift over training epochs rather than random noise; even a modest but persistent distributional shift leads to very strong statistical evidence under the KS test. More importantly for algorithm design, when we map the points of maximum separation to the time axis, the corresponding horizontal gaps reach about 650 epochs for Shoppers, 140 epochs for Default, and 40 epochs for Adult. This shows that, although the vertical gap may appear small to the eye, the practical separation in training time spans tens to hundreds of epochs. Such a margin is already sufficient to reliably distinguish the two groups during training and to support the design of our filtering strategy. We have revised the text and figures to include these quantitative measurements and to mark the separation points, making both the statistical evidence and the algorithm-relevant time-scale differences explicit.
> >
> > 5. Provide justification and comparison for algorithmic hyperparameter choices.
> >
> > In our method, the two key hyperparameters that control the behavior of DynamicCut are (i) the pruning ratio p used to select high-memorization samples and (ii) the length of the warm-up window that defines the early memorization dynamics. We already include dedicated ablation studies for these choices in the paper and further extend this analysis to strengthen the robustness discussion.
> >
> > First, for the pruning ratio p, Appendix B.1 and Table 6 compare a wide range of settings (1%, 3%, 5%, 10%, and 20%). The results show that while the exact trade-off varies with p, the overall trend is stable: very small values are too conservative and lead to limited memorization reduction, while overly large values become less precise and yield weaker or less stable improvements. A moderate choice (10%) consistently achieves the best balance between memorization reduction and utility across datasets. This shows that the effect of DynamicCut is not tied to a single fragile choice of p.
> >
> > Second, regarding the warm-up window, we treat it as a data-dependent design choice rather than a fixed universal hyperparameter, since different datasets can exhibit different training dynamics. To make this explicit, we have added a dedicated ablation study in Appendix B.2 and Table 7, where we vary the warm-up window length (e.g., 600/700/800 for Default and 2000/2100/2200 for Shoppers) and evaluate the resulting changes in both memorization mitigation and generation quality. These results show that the optimal window length is indeed dataset-dependent, and that using an overly long window tends to weaken memorization reduction. At the same time, the performance remains stable under reasonable variations of the window length, indicating that DynamicCut is not sensitive to a particular tuned choice and is robust in practice.
> >
> > 6. Revise Figure 1 to ensure fair comparison.
> >
> > We have revised Figure 1 and updated the axis ranges to be consistent and appropriately scaled across metrics. The new axes better reflect the actual magnitude of the differences and make the comparisons easier to see. We would also like to clarify that in the previous version, all axes were clearly labeled and the scales were explicitly shown, and the wider ranges were chosen mainly to keep multiple curves readable in a single plot, not to hide or distort the differences. With the revised figure, we now keep both clarity and a more faithful visual presentation of the performance gaps.

---

> > > ### Author Response · Authors · 2026-02-17
> > > **Response to Reviewer Eg4g**
> > >
> > > 7. More thoroughly analyze the trade-off between memorization and performance.
> > >
> > > The specific case pointed out, where DCM under the TabDDPM backbone on the Default dataset shows a lower alpha-Precision than TCM and TCMP in Table 3, is indeed an edge case. This behavior does not appear for the other backbones such as CTGAN and TVAE. To clarify this, we have added a more thorough analysis that aggregates results both across backbones on the same dataset and across datasets and backbones.
> > >
> > > First, when averaging over the three backbones on the Default dataset, as shown in the newly added B.9 and Table 11, DCM still achieves the strongest memorization reduction (8.78%) and at the same time attains the best MLE and the highest Trend score, while remaining comparable to TCM and TCMP on the other metrics. This shows that the single-backbone drop you observed does not reflect the overall behavior on this dataset.
> > >
> > > Second, we further add an overall summary in Table 11 that averages results across all datasets and backbones. Under this more global view, DCM achieves a substantially larger memorization reduction (13.47%) compared to TCM (8.30%) and TCMP (7.55%), while maintaining comparable generation quality. In addition, DCM reaches the best or tied-best results on several metrics, including MLE, alpha-Precision, Trend, and C2ST, and stays close to the strongest baseline on the remaining metrics.
> > >
> > > Together, these two levels of aggregation directly address the concern about the trade-off. While a small degradation can appear in a specific metric under a specific backbone, the overall results show that, in most cases, stronger memorization mitigation with DCM does not come with a clear or systematic loss in generation performance, and in some aspects even leads to improvements.

---

> > > > ### Comment · Reviewer_Eg4g · 2026-02-24
> > > >
> > > > Thanks to the authors for the clarification and revision. I have no more problems.

---

### Decision · Action_Editor_hw9o · 2026-03-22

**Recommendation:** Accept as is

**Additional Comments:**

While I recommend Accept as is, there are some formatting issues in the paper. For instance, the format of a citation is missing brackets in places where they should be. I would encourage the authors to review the paper carefully and correct the formatting errors.

**Audience:**

Yes

**Audience Explanation:**

Yes, tabular data and tabular diffusion models are of interest. Memorization is also an interesting topic. Reviewers have found the topic timely.

**Claims And Evidence:**

Yes

**Claims Explanation:**

The paper studies the problem of memorization in tabular diffusion models and proposes a new method called "DynamicCut" to mitigate memorization efficiently. Empirical evaluations are performed to analyze the performance of DynamicCut in reducing memorization, and also compared with traditional data augmentation techniques. The three reviewers have appreciated the contributions and recommended acceptance.

A few limitations noted by reviewers:
(1) limited theoretical insights with more empirical focus.
(2) benefits in terms of memorization but not so much in terms of other metrics, pointing towards a potential trade-off analysis.